# Comment on Marques et al. (2018), Channel flow, tectonic overpressure, and exhumation of high-pressure rocks in the Greater Himalayas

John P. Platt [1]

[1] Department of Earth Sciences, University of Southern California, Los Angeles, CA 90089-0740, USA

*Correspondence to*: John Platt (jplatt@usc.edu)

**Abstract.** The upward-tapering channel model proposed by Marques et al (2018) for the Himalayas has a "base" that forms part of the subducting footwall, and therefore does not close the channel. This configuration does not produce return flow, and no dynamic overpressure develops in the channel. The geometrical and kinematic configuration they actually use for

their calculations differs from this, and is both geologically and mechanically improbable. In addition, the fixed upper boundary condition in their models is mechanically unrealistic, and inconsistent with geological and geophysical constraints from the Himalayan orogen. In reality, the dynamic pressures calculated from their model, which exceed lithostatic pressure by as much as 1.5 GPa, would cause elastic flexure or permanent deformation of the upper plate. I estimate that a flexural upwarp of 50 km of the upper plate would be required to balance forces, which would lead to geologically unrealistic

topographic and gravity anomalies. The magnitude of the dynamic overpressure that could be confined is in fact limited by the shear strength of the upper plate in the Himalayas, which is likely to be <120 MPa.

*Introduction*

Marques et al (2018) (henceforth M2018) make a valuable contribution to the study of orogenic dynamics by high-lighting

the role of dynamic pressure associated with return flow in subduction channels. They calculate dynamic pressures that exceed lithostatic by 1.5 GPa or more in a large part of the channel, and suggest that the depths of metamorphism inferred from petrological data for Himalayan eclogites may therefore be overestimated by a factor of two. Before launching on this discussion, we need a couple of definitions. I will refer to the material in the subduction channel as a fluid, but we should bear in mind that in reality it is likely to be solid rock, deforming by some type of non-Newtonian creep. Second, I will use

dynamic overpressure to refer to the difference $\Delta P$ between the dynamic pressure in the fluid and the lithostatic pressure $P_L$ exerted by the weight of the overlying rock. $P_L = \int \rho(z)g\,dz$, where $\rho$ is density, and $z$ is depth. Note that dynamic overpressure as used here is generated by viscous flow in the channel, and differs in this respect from the more widely recognized concept of tectonic overpressure, which is related directly to deviatoric stress, and can exist in a static situation, with or without deformation (Schmalholz et al., 2014; Gerya, 2015).

Return flow in subduction channels has long been proposed as a mechanism for exhuming high-pressure metamorphic rocks from deep in the subduction zone. Possible drivers are buoyancy (e.g., England & Holland, 1979; Beaumont et al., 2009), topographic gradients (e.g., Beaumont et al., 2001), or dynamic overpressure (e.g., Cloos, 1982; Gerya & Stockhert, 2002). The first two mechanisms do not require the channel to be closed, but dynamic overpressure is most likely to develop if the subduction zone is closed at depth (Gerya, 2015).  This can occur where the subducting slab meets the upper plate, so that downward flow in the subduction channel is prevented, and the fluid is forced back up along the upper side of the subduction channel (Panel A in Figure 1). This phenomenon is known in the fluid-mechanics community as corner flow. Corner flow is also thought to occur in the mantle wedge above the subducting slab (e.g., Spiegelman & McKenzie, 1987). Here the symmetry is reversed, and $\Delta P$ in the corner is negative, so that asthenospheric mantle flows from the back-arc towards the corner.

Corner flow can be analyzed by solving the Napier-Stokes equations for creeping incompressible flow: $-\nabla p + \mu \nabla^2 \mathbf{v} + \rho \mathbf{g} = 0$.  These relate the spatial gradient in pressure ($p$) to the Laplacian of the velocity ($\mathbf{v}$) and the body force in the viscous channel ($\mu$ is viscosity, $\mathbf{g}$ is gravitational acceleration). The Laplacian, which comprises the second derivatives of velocity, is directly related to the stress gradients in the stress equilibrium equations, from which Navier-Stokes is derived. In a subduction channel the viscous fluid is entrained by the down-going slab, but if the upper and lower plates converge, so as to close the channel, fluid is forced away from the slab at the resulting corner (indicated by the red dot in panels A and C in Figure 1). As a result, it experiences an abrupt change in stress, and the resulting steep stress gradients require correspondingly steep pressure gradients, as shown by Navier-Stokes. The pressure gradients result in a build-up of pressure near the corner, and this in turn drives the return flow along the upper boundary of the channel. Navier-Stokes does not predict unique solutions: the dynamic overpressure is limited by the ability of the channel walls to contain it.  If the walls deform, the pattern of flow will change, and the dynamic overpressure is likely to decrease.

The analysis by M2018 suffers from some serious problems, which largely undermine their conclusions. These problems are first, there is a clear conflict between the geological configuration they use to justify their model, and the configuration they actually use. The second problem is that they assume a fixed upper boundary to the subduction channel, which cannot be defended in geological terms, and leads to unrealistic conclusions. These problems are discussed in more detail below.

*Geological configuration*

M2018 base their model on the present-day Himalayan orogen, which they interpret in terms of a subduction channel with a trapezoidal geometry produced by an irregular footwall, with features that they describe in terms of a ramp and flat geometry, as illustrated in Figure 1 of their paper.  M2018 regard the channel as being closed off by a "base" (see panel B in Figure 1 of this paper), which is clearly part of the footwall. The base is therefore part of the down-going Indian plate, and will move with the footwall at least as fast as the fluid in the subduction channel. The resulting configuration is transient; the

base will not obstruct the downward flow of the fluid, and will therefore not lead to return flow. The fluid will move down along with the footwall and the base, and because the fluid in the upper part of the channel moves more slowly than the base, $\Delta P$ will be negative where the base meets the upper plate (see panel B in Figure 1). This situation is quite different from the geometrical and kinematic configuration they actually use in the model (panel C Figure 1). Although Marques et al (2018) do not explicitly state the boundary conditions used for the base, it is clear from their model results that it is fixed with respect to the upper plate. This results in an abrupt change in the boundary conditions at the point marked with a red dot in panel C. This is the "corner" that leads to the positive dynamic overpressure and the return flow. This configuration does not resemble that in the present-day Himalaya. No present-day subduction zone has this configuration, and there is no evidence that it existed in the Himalayan subduction zone in the past. It is geologically and mechanically highly improbable, and does not provide a valid basis for statements about Himalayan orogeny or metamorphism.

*Boundary conditions*

A more fundamental problem concerns their use of a fixed upper boundary to the channel. It is true that fixed boundaries are commonly assumed in fluid mechanics problems, because the mechanical contrast between a low-viscosity fluid such as water and a steel pipe, for example, is so large that deformation of the boundaries can be neglected. In the case of the subduction channel modelled by M2018 in their Figure 2, the viscosity is 24 orders of magnitude greater than that of water, and the viscous stresses are correspondingly larger. If a dynamic overpressure of 1.5 GPa is applied from below to the upper boundary of the channel, a physical mechanism is required that is capable of keeping the boundary fixed, and M2018 give no indication what this might be. In the absence of such a mechanism, the only load acting downwards on the upper boundary is the lithostatic pressure. The forces are then unbalanced across the boundary, and Newton's laws of motion dictate that the upper plate in the Himalayas will accelerate upwards. We therefore need to discuss what mechanisms could maintain a fixed upper boundary to the channel, and whether these are geologically and mechanically reasonable.

In the real world, how can we achieve force balance on the upper boundary? The implication of a fixed boundary is that the upper plate is effectively infinitely rigid. If we accept for the moment the possibility that the upper plate is strong enough to resist permanent deformation, the upward load of 1.5 GPa will still produce an elastic response in the upper plate. An elastic plate subject to a normal load experiences an elastic deflection. The deflection produces bending moments in the plate, which counter the torque produced by the load, so the deflection increases until the load is balanced.To put this into perspective, consider the effect of the downward load of the Himalayan mountain range (5 km high on average along the crest), which amounts to ~135 MPa. It has long been established that this load produces a flexural downwarp of the underthrusting Indian plate of several km (Karner & Watts, 1983). Flexural downwarps of similar magnitude have also been documented in front of many other mountain belts, beneath ocean island volcanoes such as Hawaii, and along major transform faults (e.g., Watts & Zhong, 2000). In the case of a subduction channel, the configuration can be approximated by the analysis for flexural doming above an igneous intrusion presented by Turcotte & Schubert (2002). In this analysis, the roof of the intrusion is flexed up by magmatic pressure that exceeds lithostatic. The maximum deflection *w* is given by:

$w = \dfrac{pL^4}{384D}$, where $p$ in our case is the dynamic overpressure (total pressure less lithostatic), $L$ is the distance along the upper

plate boundary over which this pressure is applied, and $D$ is the flexural rigidity. $D$ is given by:

$D = \dfrac{Eh^3}{12(1-v)}$, where $E$ is Young's modulus, $h$ is the effective elastic thickness of the upper plate, and $v$ is Poisson's ratio. I

estimate the following values, based on Figure 2A from M2018, for the region between 40 and 100 km depth in the

subduction zone:

$L = 175$ km;

$p = 1.5$ GPa averaged over $L$. For the mechanical parameters, I have taken the following values from Jordan and Watts (2005) for the upper plate:

$E = 10^{11}$ Pa,

$h = 20$ km (Jordan and Watts give a range from $0 - 20$ km for the effective elastic thickness in southern Tibet, so I have chosen the upper limit, which minimizes the deflection),

$v = 0.25$.

The predicted deflection is 50 km: this is what is required to produce a restoring force equal to the upward load of 1.5 GPa predicted by M2018. The deflection is so large that it violates one of the assumptions of the analysis, that $w$ is small

compared to $L$. The analysis does not take into account the tapered geometry of the upper plate (which will increase the deflection), and it is sensitive to the values chosen for $E$ and $h$. But it is sufficient to demonstrate that a dynamic overpressure of 1.5 GPa in the Himalayan subduction zone is geologically unsustainable. No flexural upwarp of ~50 km amplitude has been detected in southern Tibet. To achieve a more reasonable value for the deflection (say 2 km) we would need either to choose a value of 60 km for $h$, or to reduce the dynamic overpressure to <60 MPa. A value of 60 km for the

effective elastic thickness is characteristic of the Indian plate, which is made up of granulite facies crustal rocks overlying thick and cold lithospheric mantle, but it is quite outside the range of values found for Tibet and the upper plate of the Himalayas.

*Deformable walls*

In practice, the rocks in the upper plate of the Himalayas are likely to deform permanently if subjected to significant dynamic overpressure. M2018 recognize that some permanent deformation is likely, and they attempt to address this with their deformable walls model. This section of their paper is very difficult to follow, as they do not define the thickness or geometry of the deformable walls, and their description of the boundary conditions is confusing and ambiguous. It appears that they have incorporated a layer of relatively high viscosity material into the model domain, above and below the channel.

The model domain as a whole still has fixed upper and lower boundaries, however, so the system behaves in much the same way as the model without deformable walls, and the predicted dynamic overpressure is almost identical. This model therefore fails to test the effect of deformation in the upper plate as a whole

*Permanent deformation in the upper plate*

In the real geological situation the dynamic overpressure in the channel will be limited by the brittle or plastic strength of the upper plate. Various lines of evidence suggest than an upper limit of ~120 MPa shear stress is reasonable for continental lithosphere in actively deforming regions (e.g., England & Molnar, 1991; Behr & Platt, 2014), and this is consistent with values calculated from experimental rock mechanics data (e.g., Platt & Behr, 2011, Figure 1). Cratonic lithosphere with an anhydrous feldspar-dominated lower crust can support significantly higher stresses (see Platt & Behr, 2011, Figure 2) but there is no evidence that the upper plate in the Himalayas ever had this composition. The geological evidence is that it consists of a variety of sedimentary and metamorphic rocks, minor amounts of granite, and serpentinite, and that it has a complicated internal structure, cut by abundant faults: reverse, normal and strike-slip. This is supported by the velocity structure for southern Tibet, which is typical of mid-crustal rocks (e.g., granite and metamorphic rocks with hydrous mineral assemblages) (Haines et al., 2003). Differential stresses inferred from dynamically recrystallized grain sizes in quartz range up to 28 MPa near the Main Central thrust (Law et al., 2013), and 47 MPa close to the South Tibetan detachment (Waters et al., 2018). The thermal gradient is high, and the lower part of the very thick crust in this region is likely to be close to the solidus. Wet (but solid) granitic rocks at 750°C deform readily at a differential stress of 1 MPa, with an effective viscosity ~$2 \times 10^{18}$ Pa s (Platt 2015). The values for the effective elastic thickness of the lithosphere calculated by Jordan & Watts (2005) imply that the lithosphere as a whole is unable to sustain loads of more than a few tens of MPa. A full analysis of the response of the upper plate is beyond the scope of this discussion, but it is unlikely that it could confine a dynamic overpressure in the channel greater than the shear strength of the material (Schmalholz et al., 2014). The channel and upper plate will therefore deform and change shape, invalidating the model geometry used by M2018, modifying the pattern of flow in the channel, and reducing the dynamic overpressure.

The geological and geophysical evidence therefore suggests that the upper plate of the Himalayan orogen lacks the strength required to confine dynamic overpressure with the magnitude and spatial distribution calculated by M2018. The observable limits on the both the elastic and permanent deformational responses suggest that their calculated values of dynamic overpressure are substantially too high, and do not justify the conclusions they draw about the depths at which Himalayan eclogites were metamorphosed.

*Concluding remarks*

The problems I have identified with this study raise questions about the purpose and methodology of this type of modeling. A good model is a simplified representation of the real world, allowing calculations that approximate the more complex response of the real system being studied. The model should be consistent with all physical laws, and produce results that can be tested against measurements on the real system. For a model to have any applicability to the real world, the boundary conditions must correspond at least approximately to the constraints that the real world would impose. The model set-up by M2018 does not conform with these important principles. They presented their model as a calculation of the dynamic

overpressure in a real subduction channel in the Himalayas, and they draw conclusions from it about Himalayan metamorphism. Their representation of the geometry and kinematics of the subduction channel bears so little resemblance to the real system, however, that the model predictions have to be regarded as completely unreliable. In addition, the upper boundary condition for their model is geologically and mechanically unrealistic, and fails to allow for the response of the

5   upper plate to the enormous values of dynamic overpressure they predict. As a result, these values are unlikely to have any relevance to deformational or metamorphic processes in the Himalayas.

**Acknowledgements**

I thank reviewers Stefan Schmalholz and Evangelos Moulas, and Editor Taras Gerya, for encouraging me to examine my assumptions and present my arguments as rigorously and precisely as possible.

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

**Figure Caption**

Figure 1.  A) Downward tapering subduction channel illustrating a configuration that can lead to corner flow and positive dynamic overpressure ($\Delta P$). B) Geometrical and kinematic configuration of the Himalayan subduction zone as described by Marques et al. (2018). The base of the channel moves with the lower plate, and $\Delta P$ is negative.  C) Configuration
used for calculations in the model by Marques et al. (2018).  The base is attached to the upper plate.

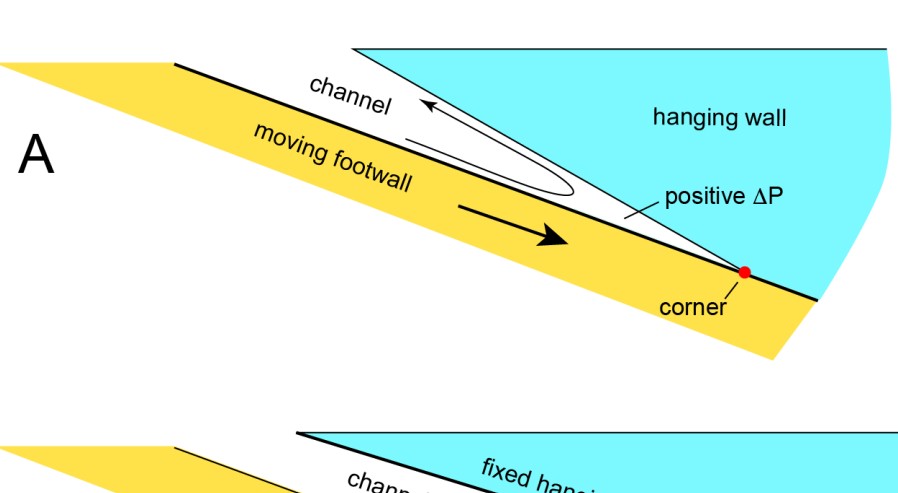

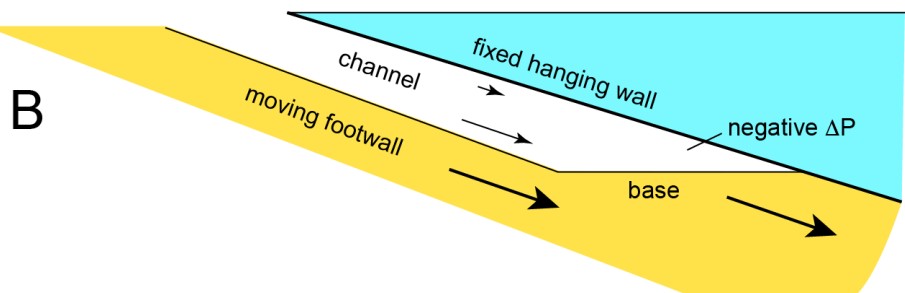

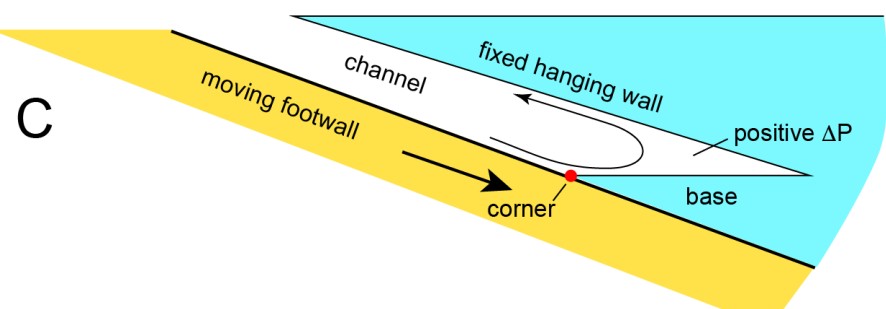

