# Peer review of "Comment on Marques et al. (2018), Channel flow, tectonic overpressure, and exhumation of high-pressure rocks in the Greater Himalayas"

_Solid Earth, 2018_

## Referee Comment (RC1) · S. M. Schmalholz (Referee) · 25 Oct 2018

Platt (2018) questions the main results of Marques et al. (2018), referred to as M2018 in the following, who use a two-dimensional numerical linear-viscous flow model to quantify magnitudes of dynamic pressure in a trapezoidal model domain. A main comment of Platt (2018) is: "I suggest that their estimates of dynamic pressure are at least one order of magnitude too high". This statement, like essentially all other statements in Platt (2018), is purely speculative and not substantiated by any mechanical calculation or alternative numerical model.

Platt (2018) argues that there are three main problems in the study of M2018:

In point 1 Platt (2018) states: "Whatever the details of the channel geometry, it must ultimately always taper downwards if it is to produce the corner-flow effect". M2018 show with their numerical simulations that return flow is generated in their model, which has an "upward-tapering" geometry. The results of M2018, hence, falsify the above statement of Platt (2018). M2018 never use the term "corner flow", but speak of "channel flow". Corner flow models commonly consider flow around a single corner. With respect to geometry, the trapezoidal model geometry of M2018 is more similar to a circulating cell model (e.g. Pollard and Fletcher, 2005; their figure 10.24).

In point 2 Platt (2018) states that the horizontal base of the model in M2018 does not move horizontally which does not fit the tectonic situation in which the Indian lower-crust and mantle lithosphere is underthrusting Tibet and hence the "base" of the Greater Himalayan Sequence should have a horizontal velocity component. This is a fair comment. However, Platt (2018) does not make any prediction about how a horizontally moving base would affect the results of M2018. In the model of M2018 there is a velocity singularity at the left edge of the model base and a model with a horizontally moving base can have a velocity singularity at the right edge of the base. The consequences of such different boundary condition have to be calculated with a corresponding numerical simulation in order to quantify the impact on the results of M2018.

Point 3: The statement of Platt (2018) that M2018 "do not allow for any motion normal to the channel boundaries" is, to the best of my knowledge, not correct. M2018 also show results for which the material above and below the channel can deform viscously. M2018 state: "This model allows for both channel walls to deform viscously, thus raising the question of how much overpressure they can retain inside the channel". Based on the description of the boundary conditions in section 3.4 of M2018, I conclude that this model allows for motion normal to the upper channel boundary.

The paragraph on page 2 from lines 6 to 18 in Platt (2018) includes mainly speculative "should-would-could" arguments, which are also mechanically unsound. For example,
Platt (2018) argues that "an unbalanced upward load of 1.5 GPa should cause a substantial flexural upwarp of the upper plate, possibly tens of km in amplitude, producing a major topographic and gravity anomaly". It is not logical why there should be an "unbalanced upward load" in a mechanical model, which is based on the equations of force balance. The dynamic pressure of 1.5 GPa is not an "unbalanced load"; this dynamic pressure and the associated pressure gradient is responsible for "pushing" the viscous material upwards, against gravity and against the downward direction of the applied boundary velocity. Platt (2018) further argues that "given that the material in the subduction channel is incompressible, even a small amount of flexural displacement would be enough to relieve the dynamic pressure". Indeed, it is well established that the dynamic pressure depends on the strength of the channel walls and dynamic pressure decreases when channel walls get weaker and, hence, displace more (e.g. Mancktelow, 2008). Such pressure relieve has been quantified with numerical models in several studies (e.g. Mancktelow, 2008) and is also mentioned in M2018 in section 1.2. M2018 report significant dynamic pressure also for models in which the viscosity of the channel was 100 to 1000 times smaller than the viscosity of the material bounding the channel (their section 3.4 and their figure 8). Moreover, the impact of elastic flexure on the dynamic pressure is not so obvious. For example, for a different model configuration for a compressed lithosphere Petrini & Podladchikov (2000) show with an analytical solution that elastic flexural loads of the upper crust and upper mantle can increase the dynamic pressure in a weak lower crust. Therefore, the elastic flexural displacement, mentioned by Platt (2018), has to be calculated with an adequate model in order to test whether and for what conditions elastic flexure causes a significant pressure relieve.

Page 2, lines 23-24. The statement of Platt (2018), "I suggest that their estimates of dynamic pressure are at least one order of magnitude too high", is not substantiated and not quantified by a mechanical calculation or model. I recommend to calculate dynamic pressure and not to suggest it.
In summary, Platt (2018) provides interesting suggestions for additional simulations, such as considering a horizontally moving model base and an elastically deformable "upper plate". However, Platt (2018) provides not a single mechanically sound calculation or alternative mechanical model, which shows that dynamic pressure magnitudes calculated by M2018 are more than one order of magnitude too high.

**Minor comments**

Page 2, Line 29: "petrologically determined depths of burial". This phrase reveals a common misunderstanding. One cannot petrologically determine a burial depth, one can only petrologically determine a thermodynamic pressure from phase equilibria calculations. To determine a burial depth, it is commonly assumed that this pressure is lithostatic. Whether pressure is lithostatic or not cannot be determined petrologically.

Page 2, 30-31: "...since the temperature determination would not be affected." Models considering energy conservation show that dissipative deformation generates heat and, hence, temperature increase. How much temperature increases has to be calculated by thermo-mechanical models.

Page 2, Line 31-33: Penniston-Dorland et al. (2015) investigate pressure to temperature ratios and argue, amongst others, that numerical subduction models are "too cold", likely because models did not incorporate all sources of heat. Since heat transfer was not calculated by M2018, comments on potential thermal results are speculative; one should perform thermo-mechanical simulations.

Best regards,

Stefan Schmalholz (P.S.: I was not reviewer of the manuscript of Marques et al. (2018))

References

Mancktelow, Neil S. "Tectonic pressure: theoretical concepts and modelled examples." Lithos 103.1-2 (2008): 149-177.
Marques, Fernando O., et al. "Channel flow, tectonic overpressure, and exhumation of high-pressure rocks in the Greater Himalayas." Solid Earth 9.5 (2018): 1061-1078.

Penniston-Dorland, Sarah C., Matthew J. Kohn, and Craig E. Manning. "The global range of subduction zone thermal structures from exhumed blueschists and eclogites: Rocks are hotter than models." Earth and Planetary Science Letters 428 (2015): 243-254.

Petrini, K., and Podladchikov, Y. "Lithospheric pressure–depth relationship in compressive regions of thickened crust." Journal of Metamorphic Geology 18 (2000): 67-77.

Platt, J. P.: Comment on Marques et al. (2018), Channel flow, tectonic overpressure, and exhumation of high-pressure rocks in the Greater Himalayas, Solid Earth Discuss., https://doi.org/10.5194/se-2018-92, in review, 2018.

Pollard, David D., and Raymond C. Fletcher. Fundamentals of structural geology. Cambridge University Press, 2005.

SED

---

## Short Comment (SC1) · 27 Oct 2018

REPLY TO PLATT'S COMMENT

F.O. Marques, N. Mandal, S. Ghosh, G. Ranalli, S. Bose

Abstract The points raised by Platt refer not to the formal correctness of our model, but rather to its relevance, given our assumptions and boundary conditions. Platt's main concern regards flexure, but his considerations, in our opinion, suffer from oversimplifications leading to unwarranted conclusions. A proper evaluation of the flexural effects of dynamic overpressure in channel flow would require a complete dynamic model in-

cluding realistic geometry and rheology, the knowledge of the elasticity and viscosity parameters, temperature, and mass transfer. At this stage, this is not available, neither to us nor to Platt. Consequently, the statement that the model results are "dramatically at variance with what we observe" is at worst unjustified, at best premature. On the contrary, the upward tapering model can explain several observations at the surface (cf. first paragraph of the Abstract in Marques et al., 2018a), and can help constrain the viscosity in the channel by keeping overpressure and outward flow within realistic values. What happens deep in a subduction zone can only be inferred, and that is why modelling is used to find possible explanations.

We thank Platt for his comments, because they give us the opportunity to clarify some critical and common questions raised when discussing tectonic/dynamic overpressure. Platt's comments reflect in great part the reasoning used by colleagues arguing against the hypothesis of high values of tectonic overpressure in subduction zones. In this reply we will contend that Platt's comments suffer from conceptual problems, and lack a quantitative analysis of the process of continental collision and tectonic overpressure development.

Regarding the three specific questions raised by Platt, we have the following considerations to offer: Upward tapering channel: It is correct to point out that the actual channel depicted in Fig. 1b of Marques et al. (2018a) tapers downward near its bottom; its overall shape, however, tapers upwards when regarded in its entirety. We therefore have chosen the shape of the model channel as shown in Fig. 1c of Marques et al. (2018a). The stated purpose of the model is to simplify geometry and rheology in order to analyse individually the effects of variations of single parameters. With the chosen boundary conditions, we obtain dynamic overpressure at relatively shallow levels with this geometry.

Boundary conditions: We state explicitly several times, including the Abstract, that no significant overpressure develops if the bottom of the channel is "open". This therefore excludes subduction channels with outflow into the sub-lithospheric mantle. As

to collision-type channels, especially when both converging plates are very "hard", the possibility of a closed base is, at least, worth exploring, especially in view of the possible consequences of overpressure for palaeodepth determinations from metamorphic peak conditions. It is true that the footwall moves down with the lower plate, but we do not see how this invalidates the model shown in Fig. 2, even if it refers only to a transient stage.

Mechanical properties of the walls: We agree that the assumption of rigid walls is an important factor affecting the results, and that the results for viscous walls are relevant only in the absence of additional external forces. The interfaces between the channel and viscous walls were mechanically coherent, and they were not kinematically constrained to restrict normal flow across the channel boundaries. Platt's claim that we did not allow for any motion normal to the channel boundaries is not correct because viscous walls must flow according to Stokes' equation, and are therefore free to deform. The inclusion of more complex dynamics would require a completely different model. We have, however, considered the case of transpression in both Marques et al. (2018a, 2018b), which has significant effects on overpressure (Fig. 7 in Marques et al., 2018a), at least in the case of rigid walls.

A major point in Platt's comment is that the possible effects of overpressure on the flexural deformation of the upper plate may invalidate the results of the model. We cannot predict what these effects could be, but we suggest that neither can Platt, at least on the basis of his comments (no quantitative analysis presented). Several parameters govern the flexural effects of vertical loads: flexural rigidity, flexural parameter, wavelength of the load, and – in the case of viscoelasticity – the age of the load (cf. e. g. Turcotte & Schubert, 2002, pp. 119-125). For wavelengths short in comparison to the thickness of the plate, the load is substantially supported by the rigidity of the lithosphere (Turcotte & Schubert, op. cit., eqn 3-111 ff.), as the bending moment required for flexural deformation would be very large. Platt used an inappropriate example, where flexural deformation occurs at a plate scale, with wavelengths far exceeding

that applicable to the present case. The geometric scale of overpressure in channel flow bears no resemblance to the scale of the whole Himalayan mountain range. In the case of overpressure from below (laccolith formation), the maximum deflection depends on the fourth power of the distance between the "pinning points" of the upper plate (Turcotte & Schubert, op. cit., eqn 3-99). Furthermore, we specifically state that the generation of overpressure is a transient process. A proper evaluation of the flexural effects of channel flow would require a complete dynamic model including realistic rheology, temperature, and mass transfer. At this stage, this is not available, neither to us nor to Platt. Consequently, the statement that the model results are "dramatically at variance with what we observe" is at worst unjustified, at best premature.

Another major point in Platt's comment is the way dynamic pressure builds up in the upward tapering channel. Platt argues (cf. Abstract of his comment) that "As a result there will be no return flow, and excess pressure will not develop in the channel . . . excess pressure is maintained by continued corner flow ". This comment gives the impression that dynamic overpressure depends on return flow or corner flow. Pressure in the Stokes' equation depends on the Laplacian of the velocity, which means that it depends on the divergence of the gradient: it can be positive, negative or zero, giving rise to underpressure, overpressure, or zero pressure, depending upon the nature of velocity gradients. The terms return flow and corner flow should be avoided, because they bear no obvious relationship to the Laplacian of the velocity, and are not the sine qua non conditions to produce dynamic pressure.

Regarding Platt's detailed comments, we have the following considerations to offer: "The upward-tapering channel model proposed by Marques et al (2018a) has a "base" that forms part of the subducting footwall, and will therefore not close the channel." (cf. first sentence of the Abstract). We cannot see how Platt reaches such a conclusion. We actually do not know what is going on down below the Himalayas, but the seismic image we used for Fig. 1b in Marques et al. (2018a) shows an overall upward tapering channel. Therefore, this is the geometry we used for the modelling of tectonic overpressure. A simplistic downward tapering channel does not portray the complexity of a continental collision zone, which can change its geometry over time and space. Furthermore, making use of Platt's reasoning, a natural downward tapering channel can leak downwards, so hampering the development of significant overpressure.

"The excess (dynamic) pressures calculated from their model, which exceed lithostatic pressure by as much as 1.5 GPa, will cause elastic flexure of the upper plate, which will relieve the excess pressure. If the excess pressure is maintained by continued corner flow, flexure of the upper plate will lead to geologically unrealistic topographic and gravity anomalies." (cf. last sentence of the Abstract). This strong statement is given without a quantitative analysis of the problem. A careful reading of the reference given by Platt (Watts & Zhong, 2000), in particular their Fig. 5, shows the effects of Maxwell's Relaxation Time and wavelength on the behaviour of a lithospheric plate. Despite citing the op. cit., Platt does not take these important parameters into account in his comments. In fact, Platt does not either take into consideration the elastic thickness of the plates in his calculations or his discussion of the rigid plates we used in the numerical model. Platt directly relates the flexural deformation with pressure drop in the channel. However, the fundamental requirement for pressure drop is by increase of the overall volume of the channel, which is difficult to quantify or even qualitatively appreciate by invoking a simple mechanical model of wall bending. The local bending or inflation the way Platt imagines in terms of ballooning may cause space accommodations locally, instead of bulk distortion of the entire lithosphere (the mechanical aspect of this phenomenon has already been discussed above). Such local adjustment will conserve the channel volume, and thereby retain the tectonic overpressure produced in the deeper section.

"A more fundamental problem arises from the assumption that the footwall and hanging wall are rigid". As we discuss in Marques et al. (2018a; cf. point 5 of section "4.3 Comparison between model and nature"), and Marques et al. (2018b), we assume rigid walls given the age of both subducting and overriding plates. See further discussion

below.

"Marques et al (2018a) try to bypass this problem by allowing viscous shear in footwall and hanging wall". We certainly did not wish to bypass any problem, on the contrary we wanted to analyse the problem of different mechanical behaviour in the bounding walls.

"...they do not allow for any motion normal to the channel boundaries.". This is not true; this comment gives the impression that Platt misread Marques et al. (2018a): (i) the shear flow partitioning, as pointed out by Platt, results not from any imposed boundary conditions at the channel walls (viscous walls do allow for any motion normal to the channel boundaries) but essentially due to the large viscosity contrast between the channel and its walls; (ii) we did allow for motion normal to the boundaries in the contractional (so-called transpressive) model (cf. Fig. 6 in Marques et al., 2018a) to investigate its effects on overpressure (cf. last paragraph of section "Boundary conditions and model set-up" and Marques et al., 2018b); (iii) the analysis of an expansional model (so-called transtensive, as in roll-back subduction) makes no sense given the existence, in the hanging wall, of the largest mountain belt and plateau on Earth.

"...all they have done is widen the channel somewhat by incorporating part of the footwall and hanging wall into it.". This is certainly not the case, because the viscosities of wall and channel are not the same. What the model shows is that three to two orders of magnitude difference in viscosity between walls (higher viscosity) and channel filling (lower viscosity) is very similar to having rigid walls when analysing tectonic overpressure.

Finally, we have never stated that proposed exhumation mechanisms are "inadequate". We wished simply to point out that the development of overpressure is a serious possibility, at least in given tectonic situations. We are aware that the reconciliation of palaeotemperature and palaeopressure estimates is a potential problem, therefore we have discussed it in Marques et al. (2018b).

We would like to take this opportunity to correct a couple of misprints in equation (A1) of our paper. The term in parentheses on the l.h.s. is the material derivative of the velocity u and should of course read $\partial u/\partial t + u\ddot{E}\H{U}\^{a}\L{G}u$; the term F on the r.h.s. is the body force per unit volume, i.e. gravity times density.

References

Marques, F.O., Mandal, N., Ghosh, S., and Ranalli, G.: Channel flow, tectonic overpressure, and exhumation of high-pressure rocks in the Greater Himalayas, Solid Earth 9, 1061-1078, 2018a.

Marques, F. O., Ranalli, G., and Mandal, N.: Tectonic overpressure at shallow depth in the lithosphere: The effects of boundary conditions, Tectonophysics, https://doi.org/10.1016/j.tecto.2018.03.022, in press, 2018b.

Turcotte D. L, Schubert, G. (2002). Geodynamics, 2nd ed., Cambridge University Press, 456pp.

Watts, A. B., and Zhong, S.: Observations of flexure and the rheology of oceanic lithosphere, Geophysical Journal International, 142, 855-875, 2000.

---

## Author Comment (AC1) · 29 Oct 2018

There are two aspects to numerical calculations like those presented by Marques et al (2018), referred to henceforth as M2018. One is the overall configuration and boundary conditions, the other is the actual calculations. The purpose of my comment was to point out that the configuration they suggest makes no geological sense, that the boundary conditions are non-physical, and that the conclusions are geologically unreasonable. Most of the points I raised should be obvious to anyone with an understanding of basic physical and geological principles. I will address them using the same structure as Schmalholz.

[Figure]

Point 1. In my comment on M2018, I pointed out that although the subduction channel they propose has a complicated geometry, it is closed at the base, and open at the top. It is therefore misleading to describe it as upward-tapering. I stand by this comment; the results of M2018 do not falsify it. There are several processes that can drive return flow in a subduction channel: buoyancy, topographic gradients, and dynamic pressure being the most widely recognized. Dynamic pressure results when a viscous fluid that is entrained by a moving boundary is forced away from the boundary and to flow in a different direction. In the subduction environment we are considering, this occurs when the subducting slab meets the upper plate, so that downward flow in the subduction channel is prevented, and the fluid is forced back up the channel by the resulting dynamic pressure. This process has been recognized and described as corner flow for more than a century. I don't know why M2018 and Schmalholz choose not to use this widely accepted term.

Point 2. Schmalholz mis-quotes me. I stated that the footwall flat, which M2018 suggest acts as the base of the channel, does move; it moves with the footwall, down the dip of the subduction zone, and will move at least as fast as the fluid near the base of the channel. Hence it cannot block the flow, and the channel will not be closed at the base. The base of the channel proposed by M2018 cannot be a footwall flat: it has to be fixed to the upper plate, and from a geological perspective it is has a highly improbable geometry. A numerical simulation isn't required to demonstrate this problem, but a basic knowledge of structural geology is helpful.

Point 3. Schmalholz disputes my statement that the models in M2018 do not allow for motion normal to the channel boundaries. M2018 clearly state that the upper plate is rigid and fixed; the lower plate can only move parallel to the channel boundary. For the model with deformable walls, they "chose a geometry with kinematic boundary conditions as in the reference model with rigid walls" (line 305-6 in M2018). The thicknesses of the deformable walls are not given, but they appear to be indicated in Figure 8A. The interfaces between the deformable walls and the rigid plates above and below are given a no-slip boundary condition. This effectively constrains the flow in the deformable walls to be parallel to the rigid bounding plates.

Schmalholz also disputes my statement that the dynamic pressure in the models produces an unbalanced load on the upper plate. He is wrong. M2018 used the Navier-Stokes equations for creeping flow in the channel, which are based on the stress equilibrium equations, so the forces in the channel are in equilibrium. But the boundary conditions they chose are non-physical. The loads normal to the upper boundary of the channel consist of the pressure in the channel (lithostatic load + dynamic pressure) on one side, and the lithostatic load alone on the other side. In M2018 they try to justify this by stating that the upper plate is rigid. Even a rigid plate will exhibit elastic behavior, however, and it has been well known for more than 150 years that an unbalanced load normal to an elastic plate causes a flexural deflection. The deflection is resisted by bending moments in the plate, which increase with the deflection until the load is balanced. The resistance scales with the deflection: if there is no deflection, there is no resistance. Hence in the model proposed by M2018, where no motion normal to the boundaries is allowed, the forces will be unbalanced. My statement was neither speculative nor mechanically unsound.

Schmalholz, after some irrelevant discussion, states that the elastic flexure of the upper plate has to be calculated with an adequate model. I show here with an approximate analytical calculation that modeling is not required. The configuration suggested by M2018 can be approximated by the analysis for flexural doming above an igneous intrusion presented by Turcotte & Schubert (2002). In this analysis, the roof of the intrusion is flexed up by magmatic pressure that exceeds lithostatic. A separate file with the analysis is attached as a supplement. The predicted deflection is 50 km. This is so large that it violates one of the assumptions of the analysis, that the deflection is small compared to the distance over which the excess pressure is applied. The analysis also does not take into account the tapering geometry of the upper plate, and it is sensitive to the values chosen for the elastic modulus and the effective elastic thickness. But it is

sufficient to demonstrate that a dynamic pressure of 1.5 GPa in the Himalayan subduction zone is geologically unsustainable. No flexural upwarp of ∼50 km amplitude has been detected in southern Tibet. A full numerical analysis of the flexure is unwarranted at this point. What is needed is for the authors of M2018 to reconsider their modeling configuration, remove the non-physical boundary conditions, and come up with a revised model (including a deformable upper plate) that is consistent with geological constraints.

I suggested that the dynamic pressures estimated by M2018 are likely to be too high by at least an order of magnitude. This was based on the flexural effect discussed in the previous paragraph. Given that there are various free parameters in their model (e.g., the viscosity), I thought it would be helpful to indicate the range in which a geologically acceptable result might fall. It is not incumbent on me to reconfigure their model and carry out a numerical exercise: that's a job for the authors. I'm interested in the results, but this type of modeling is not my area of expertise.

Schmalholz's minor comments. "petrologically determined depths of burial". Perhaps I should have written petrologically inferred depths of burial. This doesn't affect the point I was making.

"the temperature determination would not be affected". Changing the petrologically determined depth of burial, without changing the pressure, does not affect the petrologically determined temperature, regardless of the heat sources for metamorphism. My point was that postulating a decreased depth of burial for UHP rocks exacerbates what has been identified as a potential problem with thermal gradients in subduction zones.

Jordan, T. A., and Watts, A. B., 2005, Gravity anomalies, flexure and the elastic thickness structure of the India-Eurasia collisional system: Earth and Planetary Science Letters, v. 236, p. 732-750 Marques, F.O., Mandal, N., Ghosh, S., and Ranalli, G ., Channel flow, tectonic overpressure, and exhumation of high-pressure rocks in

the Greater Himalayas. Solid Earth 9.5 (2018): 1061-1078. Turcotte D. L, Schubert, G. (2002). Geodynamics, 2nd ed., Cambridge University Press, 456pp.

Please also note the supplement to this comment:
https://www.solid-earth-discuss.net/se-2018-92/se-2018-92-AC1-supplement.pdf

**Supplement:**

The configuration suggested by M2018 can be approximated by the analysis for flexural doming above an igneous intrusion presented by Turcotte & Schubert (2002). In this analysis, the roof of the intrusion is flexed up by magmatic pressure that exceeds lithostatic. The maximum deflection $w$ is given by:

$$w = \frac{pL^4}{384D},$$ where $p$ in our case is the dynamic pressure (total pressure less lithostatic), $L$ is the distance along the upper plate boundary over which this pressure is applied, and $D$ is the flexural rigidity. $D$ is given by:

$$D = \frac{Eh^3}{12(1-v^2)},$$ where $E$ is Young's modulus, $h$ is the effective elastic thickness of the upper plate, and $v$ is Poisson's ratio. I estimate the following values, based on Figure 2A from M2018, for the region between 40 and 100 km depth in the subduction zone.

$L$ = 175 km;

$p$ = 1.5 GPa averaged over $L$.

For the mechanical parameters, I have taken the following values from Jordan and Watts (2005) for the upper plate:

$E = 10^{11}$ Pa,

$h$ = 20 km (Jordan and Watts give a range from 0 – 20 km for the effective elastic thickness in Tibet, so I have taken a conservative value),

$v = 0.25$.

The predicted deflection is 50 km.

---

## Author Comment (AC2) · 5 Nov 2018

The Reply by Marques et al. to my Comment on their paper suggests that some of my criticisms were insufficiently clear and precise. I take the opportunity here to clarify the most important points with the help of some diagrams and a simple mechanical analysis.

Firstly, there is a clear conflict between the geological configuration they use to justify their model, and the configuration they actually use. The geological situation, based on the present-day Himalayan orogen, involves an irregular footwall with features that they describe in terms of a ramp and flat geometry (panel A in the figure, attached as a

supplement). These features are part of the down-going Indian plate, and hence move with the footwall at least as fast as the material in the subduction channel. The footwall flat, which they describe as the "base" to the channel, will not obstruct the downward flow of the material in the channel, and will therefore not lead to return flow. The material in the channel will simply move down along with the footwall and the base, and the dynamic pressure will in fact be negative where the base meets the upper plate. This is quite different from the geometrical and kinematic configuration they use in the model (panel B in the figure). Although Marques et al (2018) do not explicitly state the boundary conditions used for the base, it is clear from their model results that it is fixed with respect to the upper plate. This results in an abrupt change in the boundary conditions at the point marked with a red dot in panel B. This is the "corner" that leads to the positive dynamic pressure and the return flow. This configuration does not resemble that in the present-day Himalaya, and is geologically highly improbable. No present-day subduction zone has this configuration, and there is no evidence that it existed in the Himalayan subduction zone in the past. It would have been much more useful had they used a simple downward tapering geometry for the channel (e.g., panel C in the figure), and calculated the resulting dynamic pressure as a function of viscosity and rate of subduction. This would be a real contribution to the more general problem of return flow in subduction zones, though it would have to be done together with a realistic analysis of the response of the upper plate (see next point).

My second criticism concerns their use of a fixed upper boundary to the channel. The kinematic boundary condition they specify is essentially non-physical, because it results in unbalanced forces across it: the normal load on one side is the total pressure in the channel (lithostatic plus dynamic pressure), and on the other it is just the lithostatic load. It is true that rigid boundaries are commonly used in fluid mechanics problems, because the mechanical contrast between a low-viscosity fluid such as water and a steel pipe, for example, is so large that deformation of the boundaries can be neglected. In the case of a subduction channel, however, the modeled viscosity is more than 20 orders of magnitude greater than that of water, and the viscous stresses are

correspondingly larger. No realistic geological material can withstand an unbalanced normal load of 1.5 GPa without either significant elastic flexure or permanent deformation. The upper plate in the Himalayas consists of thick continental crust with a complex internal structure, and is not particularly strong. Unless it deforms permanently (which is likely), it will flex until the bending moments are large enough to balance the enormous load. The laccolith analysis from Turcotte & Schubert (2002) provides a good enough approximation to the situation – it's a pity Marques et al. (2018) didn't try this calculation before publishing their paper. Using realistic parameters, it predicts a flexural upwarp of 50 km (see supplement), which is so absurdly large that it demonstrates unequivocally how unrealistic their predictions of dynamic pressure are in geological terms.

The "deformable walls" model described in Marques et al (2018) does not change this conclusion: they still specify a fixed upper plate; this, and the no-slip boundary condition between the deforming walls and the upper plate, restrict deformation in the walls to shear parallel to the boundaries. Hence the unbalanced force condition across the boundary with the fixed upper plate remains.

It seems logical that the dynamic pressure will in practice be limited by the strength of the upper plate. Various lines of evidence suggest than an upper limit of ∼120 MPa shear stress is reasonable for continental lithosphere in actively deforming regions (e.g., England & Molnar, 1991; Behr & Platt, 2014), and this is consistent with values calculated from experimental rock mechanics data (e.g., Platt & Behr, 2011). This limits the stresses associated with the flexural response to the dynamic pressure, and hence could be used to place an upper bound on the magnitude of the dynamic pressure.

The remarks about the causes of dynamic pressure and return flow in the Reply serve only to obscure the underlying physics of the situation, so some clarification is needed. The Navier-Stokes equations relate the pressure gradient to the Laplacian of the velocity and the body force in the viscous channel. The Laplacian, which comprises second derivatives of velocity, is directly related to the stress gradients in the stress equilibrium

equations, from which Navier-Stokes is derived. In a subduction channel viscous material is entrained by the down-going slab, but if the upper and lower plates converge, so as to close the channel, this material is forced away from the slab at the resulting corner (indicated by the red dot in panel C of the figure). As a result it experiences an abrupt change in stress, and the resulting steep stress gradients require correspondingly steep pressure gradients, as shown by Navier-Stokes. The pressure gradients result in a build-up of pressure near the corner, and this in turn drives the return flow along the upper boundary of the channel. That's how corner flow works, and that's why it's called corner flow.

Behr, W. M., and Platt, J. P., 2014, Brittle faults are weak, yet the ductile middle crust is strong: Implications for lithospheric mechanics: Geophysical Research Letters, v. 41, p. 8067-8075. doi:10.1002/2014GL061349.

England, P. C., and Molnar, P., 1991, Inferences of deviatoric stress in actively deforming belts from simple physical models: Royal Society of London Philosophical Transactions, v. A337, p. 151-164.

Marques, F.O., Mandal, N., Ghosh, S., and Ranalli, G ., Channel flow, tectonic overpressure, and exhumation of high-pressure rocks in the Greater Himalayas. Solid Earth 9.5 (2018): 1061-1078.

Platt, J. P., and Behr, W. M., 2011, Deep structure of lithospheric fault zones: Geophysical Research Letters, v. 38, p. L24308. doi:10.1029/2011GL049719.

Turcotte D. L, Schubert, G. (2002). Geodynamics, 2nd ed., Cambridge University Press, 456pp.

Please also note the supplement to this comment:
https://www.solid-earth-discuss.net/se-2018-92/se-2018-92-AC2-supplement.pdf

[Figure]

[Figure]

**Fig. 1.** A) Configuration of the Himalayan subduction zone described by Marques et al. B) Configuration used in the model. C) Standard configuration for corner flow in a subduction zone.

---

## Referee Comment (RC2) · E. Moulas (Referee) · 21 Nov 2018

The comment posted by Prof. Platt (hereafter P18) highlights some points of the model proposed by Marques and co-workers (hereafter M18) in their publication entitled "Channel flow, tectonic overpressure, and exhumation of high-pressure rocks in the Greater Himalayas" in a very critical manner. To the author's opinion, the most essential criticism of P18 on M18 model is the model configuration. Based on P18, the contact between the subduction channel and its overriding plate, as presented in the model of M18, is "unphysical" and it leads to erroneous predictions of tectonic overpressure (TOP). The characterization "unphysical" and the subsequent arguments

used by P18 are based on erroneous assumptions that lead to unphysical conclusions, and are therefore unjustified. Before I go into the details of the comment of P18 I will highlight the main aspects of the model configuration presented by M18. The models made by M18 can be broadly separated into 2 categories A. Models that concern only the channel. B. Models that consider a broader model domain where the channel and its overriding plate are deformable (section "Viscous deformable walls" p.1067-1068 in M18). The first category of models was criticized by P18 on the basis that, in order to have such velocities, the material of the overriding plate must be undeformed. However, for the second category of models shown by M18, the authors clearly state that they have relaxed this assumption and they actually considered deformable walls. In addition, M18 suggest that if the viscosity of the channel walls is at least 100 times larger than that of the channel then TOP can be significant and the overriding plate remains essentially undeformed. However, M18 do not provide details on how "significant" is the magnitude of TOP in that case. In the case where the viscosity of the channel is 1000 times larger than that of the walls then TOP is in the order of a GPa and then clearly the authors state that their conclusions regarding the TOP depend on the strength of the walls. P18 argues that if there is a significant amount of TOP, then, the "excess" pressure will cause elastic flexure of the upper plate that would be unrealistic. Following P18, the deflection of the overriding plate is a consequence of having "unbalanced loads" in the channel. This criticism by P18 reveals a misconception of P18 regarding force balance in Stokes' equations. The model of M18 satisfies force balance everywhere within the model. One cannot make predictions of the magnitude of the applied stresses in regions outside the model domain. For example, in a follow-up comment (29-Oct-2018), Prof. Platt argued that the TOP in the viscous channel is unbalanced since the overriding plate experiences lithostatic pressure. The last statement clearly reveals a mechanical misconception, i.e. it was the implicit assumption of P18 that pressure is lithostatic in the overriding plate. The last statement is mechanically unfeasible and violates force balance. In other words, there is no moment in time where there would have been significant TOP in the channel and lithostatic stress in

the channel wall. Therefore, if there is a significant TOP in the channel then one needs to solve for the state of stress outside of the channel in order to have any meaningful stress estimates. In summary, the large values of TOP predicted by M18 are just the outcome of model inputs regarding the geometry, the specific rheology, the overall boundary conditions etc. How appropriate are these estimates needs to be verified and quantified. Without specific information on the stress distribution on the models of M18, one cannot judge how realistic these results are with respect to their stress magnitudes and the strength that they imply for the overriding plate. The plot given by M18 regarding TOP (their figure 8b) is not sufficient to draw meaningful conclusions for the state of stress in the channel and its bounding wall. In addition, when large regions of the lithosphere are considered, the pressure and temperature changes that occur within the lithosphere will have a large effect on the active deformation mechanisms and rheological parameters. Consequently, the choice of the rheological behavior and the geometrical configuration will have a first-order control on the stress predictions.

Specific comments P1-l.15-25, M18 did not state that they have a typical corner-flow model. Therefore, there is no justification for the suggestions of P18 regarding the tapering angles of the models of M18. P1-l.28 ". . .at the same velocity as the downward flow in the channel" There is no specific reason on why the downward velocity must be exactly the same as the one of the plate especially when rheological boundaries are considered. Perhaps, having it perfectly immobile like in the case of M18 may be an exaggeration. However, without having a self-consistent model where this channel would be an emerging rather than an imposed feature, no quantitative estimates can be given. P2-l.3-5 In their paper, M18 clearly state that they consider also the case of deformable walls therefore all this section is not justified. P2.l.6-24 All this part is speculative and based on erroneous implicit assumptions. i.e. there is no reason why the load must be unbalanced. Therefore, all the arguments that follow (e.g. about unrealistically large flexures etc.) are based on faulty assumptions. Additionally, the question posed by P18 on "why are the predictions of M18 so dramatically at variance with what we observe?" is misleading since the "predictions" were made by P18 and not by M18. P2.29-33 The

statement "petrologically determined depths" by P18 confirms the author's view about mechanical misconceptions in the arguments used by P18. I would like to highlight that the petrologically "determined" (or inferred) depths are usually a result of the lithostatic pressure formula. The lithostatic pressure formula can be derived from Stokes' equations if one assumes that there are no differential stresses, no topography and that density is stratified e.g. (Gerya, 2015; Moulas et al., 2018). The pressure is therefore an outcome of a mechanical model and not an independent constraint.

References Gerya, T., 2015. Tectonic overpressure and underpressure in lithospheric tectonics and metamorphism. Journal of Metamorphic Geology 33, 785–800. https://doi.org/10.1111/jmg.12144 Moulas, E., Schmalholz, S.M., Podladchikov, Y., Tajčmanová, L., Kostopoulos, D., Baumgartner, L., 2018. Relation between mean stress, thermodynamic and lithostatic pressure. Journal of Metamorphic Geology in press.

---

## Referee Comment (RC3) · Schmalholz (Referee) · 26 Nov 2018

Platt commented on my review of his original comment on Marques et al. (2018), referred to as M2018. I disagree with the arguments of Platt, but I focus here only on the main points.

M2018 use a model with kinematic boundary conditions in which they set the velocities of the upper channel wall to zero. Platt argues that this boundary condition is non-physical. Actually, most corner and channel flow models applied in the Earth Sciences (e.g. flow in a mantle wedge, flow mélanges or return flow in a subduction channel; e.g. Cloos, 1982; England and Holland, 1979; Turcotte and Schubert, 2014) are based on such kinematic boundary condition. Usually, a corner/channel-parallel velocity is applied on one side and on the other side, typically the hanging wall, the velocities are zero. This boundary condition of zero velocity and the mechanical constraint of force balance require that the loads normal to the upper channel boundary, which are exerted by the channel, must be equal to the loads normal to the upper channel boundary exerted by the hanging wall. However, Platt argues that "The loads normal to the upper boundary of the channel consist of the pressure in the channel (lithostatic load + dynamic pressure) on one side, and the lithostatic load alone on the other side". This statement of Platt is wrong because it violates the force balance across the upper channel wall. Force balance requires that the load (more precisely the total stress normal to the boundary) of the hanging wall must balance the load of the channel (for slow deformation if inertial forces are neglected). It is, hence, not the boundary condition of M2018, which is non-physical, it is the assumption of Platt of a lithostatic load in the hanging wall which is non-physical in the context of the model of M2018. Therefore, I argue that Platt's argument is mechanically not sound, because the stress state he assumes violates the force balance across the upper channel wall.

I explain the force balance between channel and hanging wall for illustrative purposes with the simple elastic flexure model used by Platt. This flexural model quantifies the maximal deflection of an elastic beam, mimicking the resistance of the hanging wall, under the action of an external pressure, representing the dynamic pressure in the channel, acting normal to the initially straight and unstressed beam (Turcotte and Schubert, 2014). The maximal deflection of the beam is

$$w_{max} = \frac{L^4 p}{D \, 384} = \frac{L^4}{h^3} \frac{p(1-v^2)}{E} \frac{12}{384} \qquad (1)$$

where L is the length of the beam, p is the external pressure, D is the flexural rigidity of the beam, h is the effective elastic thickness of the beam, E is Young's modulus and v is Poisson's ratio. Platt used L = 175 km, p = 1.5 GPa, E = $10^{11}$ Pa and v = 0.25, and he assumed for the effective elastic thickness (Burov and Diament, 1995) of the beam a value of h = 20 km so that he obtained $w_{max} \approx 50$ km (Figure 1A). Obviously, Platt's assumption of h = 20 km disagrees with the assumption of M2018 who assumed that the hanging wall is strong enough so that the deflection of the hanging wall under the action of a dynamic pressure is small compared to the length of the channel wall. Equation (1) shows the strong non-linear dependence of $w_{max}$ on both L and h. If one uses h = 60 km then $w_{max} < 2$ km (Figure 1A). In Figure 1B I plot the corresponding deflected geometry of the beam for h = 20, 40 and 60 km using also the solution from Turcotte and Schubert (2014):

$$w(x) = w_{max}\left(1 - 8\frac{x^2}{L^2} + 16\frac{x^4}{L^4}\right) \qquad (2)$$

where x is the coordinate along the beam. For h = 60 km the deflection of ca. 2 km is very small compared to the length of the beam of 175 km (ca. two orders of magnitude). A deflected, and hence stressed, elastic beam with h = ca. 60 km and with a deflection of ca. 2 km has deviatoric stresses

which statically balance the applied external load of p = 1.5 GPa. If one assumes that the hanging wall has sufficient strength, which would correspond for our simple example to the flexural resistance of a beam with h ≥ ca. 60 km, then one can approximate the upper channel boundary as straight and stationary, that is, the velocities are zero under the action of p = 1.5 GPa, as assumed by M2018. The mechanically correct description of the scenario considered by M2018 is a strong hanging wall with non-lithostatic stresses which statically balance the dynamic pressure of the channel. Therefore, the boundary conditions of M2018 are neither non-physical nor are the loads between channel and hanging wall unbalanced. The boundary condition of M2018 simply requires that the hanging wall is strong enough to balance the dynamic pressure with a small deflection of the channel wall. For h = 40 km the maximal deflection is ca. 6 km. How much a curved channel wall corresponding to a deflection of up to ca. 10 km affects the dynamic pressure in the channel should be quantified with corresponding 2D numerical simulations including an elastically deformable hanging wall.

[Figure]

[Figure]

Figure 1. A) Maximum deflection, $w_{max}$ (see equation 1), as function of effective elastic thickness, h. B) Variation of elastic deflection along a 175 km long beam. Maximum deflection (in A) occurs at the horizontal center (0 km).

A mechanically sound, and in my opinion justified, critical comment on the model of M2018 would be that they assume a strength of the hanging wall which might be too large for the situation in Tibet. Platt argues that estimates of dynamic pressure of M2018 are at least one order of magnitude too high, that is, dynamic pressure should be < 0.15 GPa. To test whether the order of magnitude of dynamic pressure suggested by M2018 is feasible, I consider p = 0.5 GPa, which is still a significant dynamic pressure, and reduce L = 175 km by 20% to L = 140 km (Figure 1A). The maximum deflection is then dramatically reduced (Figure 1A) and if we assume that values of h between 20 and 30 km for the hanging wall might be possible then the model of M2018 seems feasible with respect to the

strength of the hanging wall. Therefore, I argue that the strong statement of Platt, that estimates of dynamic pressure of M2018 are at least one order of magnitude too high, is not justified.

The elastic beam model is not very suitable to quantify the mechanical resistance of the hanging wall because its deformation behavior in depths larger than 40 km is likely dominated by plastic and ductile deformation. Hence, its effective strength depends on the assumed flow law (feldspar or quartz dominated, wet or dry etc.), the temperature variation in the hanging wall and its geometry. However, I agree with Platt that the elastic beam model is a good model to check, to first-order, the feasibility of channel models with high dynamic pressure. M2018 expose themselves to critics of Platt because they consider a very long channel in which high dynamic pressure acts on the upper channel boundary over a length, L, of ca. 175 km. Equation (1) shows the extreme nonlinear dependence of $w_{max}$ on L and h ( $w_{max} \sim L^4 / h^3$ ). Since I am interested whether lithospheric channel flow models in general could support dynamic pressures in the order of 1 GPa I discuss, based on equation (1), the feasible magnitudes of dynamic pressure in lithospheric channels. For example, M2018 used a channel with a base at 100 km depth but they did not perform models with a smaller channel height (the vertical distance between base and surface). If one considers Figure 1B of M2018 one could also consider a channel with a base at 70 km depth so that the upper channel wall exposed to high dynamic pressure of ca. 1 GPa would likely be ca. 85 km shorter ( 30 km / sin(20°) ). For a beam with L = 90 km (175 km – 85 km) and p = 1 GPa the maximal deflection for h = 20 km is only ca. 2 km (Figure 1A). Therefore, reducing the channel height and, hence, the length of the channel wall exposed to dynamic pressures of ca. 1 GPa causes a dramatic decrease of the potential deflection of the hanging wall. The model of an elastic beam, representing the resistance of the hanging wall, shows that dynamic pressure in the order of 1 GPa can be easily balanced by channel walls with a length of several tens of kilometer, whereby the associated deflection of the channel wall is small and the required effective elastic thickness of the hanging wall is reasonable. 2D thermo-mechanical numerical simulations of a compressed lithosphere with viscoelastoplastic deformation behavior also support this conclusion and show that dynamic pressure between 0.5 and 1 GPa can develop in weak crustal-scale shear zones and that these dynamic pressures are balanced by surrounding rock units without tens of kilometers deflection of the shear zone walls (e.g. Schmalholz and Podladchikov, 2013). Furthermore, one should keep in mind that it is well established by host-inclusion studies of minerals that pressure on the mineral scale can deviate from the lithostatic pressure by magnitudes in the order of 1.5 GPa (e.g. Angel et al., 2015; and references therein). Whether such pressure deviations occur in nature on significantly larger scales is currently disputed, because the effective strength of polymineralic rock units at depth is still contentious. However, some field- and petrology-based studies provide strong support for dynamic pressure variations between ca. 0.5 and 2 GPa in natural rock units on the outcrop scale (Chu et al., 2017; Vrijmoed et al., 2009). Recently, Jamtveit et al. (2018) argued that petrological and geochronological data and corresponding field observations of meter-scale eclogite shear zones in granulites in Western Norway are best explained if a dynamic (or tectonic) pressure of ca. 0.5 GPa is considered.

Concerning point 3 of my review whether the model of M2018 with a viscously deforming hanging wall allows motion normal to the upper channel boundary: The authors of M2018 have already replied to Platt and they confirmed my evaluation that the comment of Platt, claiming that motion normal to the channel wall was not possible, is incorrect. I trust that the authors of M2018 have checked their model and boundary condition before their reply to Platt.

References

Angel, R., Nimis, P., Mazzucchelli, M., Alvaro, M., Nestola, F., 2015. How large are departures from lithostatic pressure? Constraints from host–inclusion elasticity. Journal of Metamorphic Geology 33, 801-813.

Burov, E.B., Diament, M., 1995. The effective elastic thickness (Te) of continental lithosphere: What does it really mean? J. Geophys. Res. 100, 3905-3927.

Chu, X., Ague, J.J., Podladchikov, Y.Y., Tian, M., 2017. Ultrafast eclogite formation via melting-induced overpressure. Earth and Planetary Science Letters 479, 1-17.

Cloos, M., 1982. Flow melanges: Numerical modeling and geologic constraints on their origin in the Franciscan subduction complex, California. Geological Society of America Bulletin 93, 330-345.

England, P.C., Holland, T.J.B., 1979. Archimedes and the Tauern eclogites - role of buoyancy in the preservation of exotic eclogite blocks. Earth and Planetary Science Letters 44, 287-294.

Jamtveit, B., Moulas, E., Andersen, T.B., Austrheim, H., Corfu, F., Petley-Ragan, A., Schmalholz, S.M., 2018. High Pressure Metamorphism Caused by Fluid Induced Weakening of Deep Continental Crust. Scientific Reports 8, 17011.

Marques, F.O., Mandal, N., Ghosh, S., Ranalli, G., Bose, S., 2018. Channel flow, tectonic overpressure, and exhumation of high-pressure rocks in the Greater Himalayas. Solid Earth 9, 1061-1078.

Schmalholz, S.M., Podladchikov, Y.Y., 2013. Tectonic overpressure in weak crustal-scale shear zones and implications for the exhumation of high-pressure rocks. Geophysical Research Letters 40, 1984-1988.

Turcotte, D., Schubert, G., 2014. Geodynamics. Cambridge University Press.

Vrijmoed, J.C., Podladchikov, Y.Y., Andersen, T.B., Hartz, E.H., 2009. An alternative model for ultra-high pressure in the Svartberget Fe-Ti garnet-peridotite, Western Gneiss Region, Norway. European Journal of Mineralogy 21, 1119-1133.

---

## Author Comment (AC3) · 28 Nov 2018

Response to Review by E. Moulas J. P. Platt jplatt@usc.edu

I have already addressed many of the points raised by Moulas in my response to the Reply by Marques et al., so in the interests of brevity I will just emphasize a few of the critical issues.

Moulas dismisses my concerns about the boundary conditions, suggesting that because Marques et al. (hereafter M18) used the Navier Stokes equations to solve for the velocities in the channel, the boundary conditions will obey the requirement for

force balance. This is incorrect. The boundary conditions were not calculated using Navier-Stokes: they were imposed arbitrarily by M18. A physical process is required that is capable of keeping the boundary fixed, and M18 gave no indication what this might be. In their model set up, the only load acting on the upper boundary is the weight of the overlying rock.

Moulas's statement "One cannot make predictions of the magnitude of the applied stresses in regions outside the model domain" is an abrogation of scientific responsibility. M18 presented their model as a calculation of the dynamic pressure in a real subduction channel in the Himalayas, and they draw conclusions from it about Himalayan metamorphism. We have to consider the tectonic context of the model, and in fact M18 in their paper discuss the fact that the strength of the walls is an important factor governing the dynamic pressure.

Responses to the specific comments:

"M18 did not state that they have a typical corner-flow model". True, but it is a corner flow model: see my response to the Reply by M18. The channel has to close downward, and M18 state this.

"There is no specific reason on why the downward velocity (of the footwall ramp) must be exactly the same as the one of the plate". M18 state that the underthrust plate is rigid. The footwall ramp is part of the underthrust plate, so it must move with it. M18 make no reference to "rheological boundaries". They take a feature that they describe as part of the lower plate, and give it the same velocity as the upper plate. That isn't an exaggeration: it's simply wrong.

The remaining points have been addressed in my other posts. Note that loads generated by differential topography in the Himalayas are unlikely to exceed 135 MPa. Differential stresses are unlikely to have exceeded 200 MPa: the upper plate in the Himalayas consists of a variety of sedimentary and metamorphic rocks, minor amounts of granite, and serpentinite. It is cut by abundant faults: reverse, normal and strike-slip.

The microstructure (e.g., dynamically recrystallized grain size in quartz) suggests differential stresses up to 28 MPa (Law et al., 2013). The effective elastic thickness of the lithosphere in that region, calculated from the admittance between topography and free air gravity, is in the range 0-20 km, implying that it is unable to sustain loads of more than a few tens of MPa (Jordan & Watts (2005).

Jordan, T. A., and Watts, A. B., 2005, Gravity anomalies, flexure and the elastic thickness structure of the India-Eurasia collisional system: Earth and Planetary Science Letters, v. 236, p. 732-750.

Law et al., 2013, Deformation temperatures and flow vorticities near the base of the Greater Himalayan Series, Sutlej Valley and Shimla Klippe, NW India. Journal of Structural Geology, v. 54, p. 21-53.

---

## Author Comment (AC4) · 28 Nov 2018

Response to Comment by Stefan Schmalholz J. P. Platt jplatt@usc.edu

What a pleasure to see some genuine scientific discussion of the issues! Up to this point the discussion seems to have consisted of flat denial that there is a problem, combined with pejorative remarks about my professional competence.

Schmalholz's analysis of the magnitude of the elastic flexure of the upper plate in this situation is very useful. My only comment is that the value for L that I used was measured directly from Figure 2 of M18, being the distance along the upper boundary of

the channel over which the dynamic pressure exceeds 1.5 GPa. My value for h was taken from Jordan & Watts (2005), who determine the effective elastic thickness over the whole region from the admittance between topography and free air gravity. 20 km is the maximum possible value for southern Tibet: they give the range as 0-20 km. This provides direct observational evidence that the overriding plate is weak, and unable to sustain topographic loads of more than a few tens of MPa. To get a 60 km value for h we would need to go into the Indian shield, which is composed of dry granulite facies rocks overlying cold lithospheric mantle.

It is quite true, as Schmalholz states, that many fluid-mechanical models assume fixed boundaries. That doesn't mean this assumption is right. For low viscosity fluids such as water flowing in a steel pipe, it is a reasonable approximation. In the case of the subduction channel modeled by M18 in their Figure 2, the viscosity is 24 orders of magnitude greater than that of water, and the viscous stresses are correspondingly larger. A physical process is required that is capable of keeping the boundary fixed, and M18 gave no indication what this might be. In the absence of such a process, the only load acting downwards on the upper boundary of the model is the weight of the overlying rock. The forces are then unbalanced, which would lead to the uncomfortable conclusion that the entire upper plate in the Himalayas would accelerate upwards at ∼1g. Schmalholz helpfully provides some physical scenarios that could avert this catastrophe, involving elastic deflection of the upper plate, but M18 did not: hence my critical comment.

With respect to the model with deformable walls, I based my comment on the statement in M18 that they "chose a geometry with kinematic boundary conditions as in the reference model with rigid walls" (line 305-6 in M2018). This effectively constrains the flow in the deformable walls to be parallel to the rigid bounding plates, and this is confirmed by the velocity trajectories shown in Figure 8 of M18. In their Reply, M18 did not address this issue.

My thanks once again to Schmalholz for his constructive discussion.

---

## Referee Comment (RC4) · Moulas (Referee) · 29 Nov 2018

Prof. Platt uses an elastic flexure formula (from analytical solution) in order to calculate the corresponding deflection of the overriding plate in the models of Marques et al. (2018). This loading will instantaneously deform the overriding plate as much as 50km upward. This is a consequence of the elastic rheology that is utilized by Prof. Platt and not included in the models of Marques et al. (2018). Since the elastic response is instantaneous, therefore, the current subduction configuration in nature corresponds to the stressed state, i.e. the stress state AFTER the loading. The 50km deflection calculated by Prof. Platt could be hypothetically observed as the result of unloading

to a state with negligible tectonic overpressure (TOP). Since such deflections seem unrealistic, the hypothetical stress state where TOP would be insignificant is equally unrealistic. This stress state is expected after the gravitational collapse of the mountain ranges and their roots, i.e. towards flat Moho and topography. This stress state would therefore not be envisioned in active belts like the Himalayas.

Minor issues: Prof. Platt states that my view regarding the predictive ability of Stokes' equations outside the model domain "is an abrogation of scientific responsibility". I agree that it is the responsibility of the researchers to check the applicability of their boundary conditions in their models. As I already mentioned in my previous comment, Marques et al. considered two types of models. 1) A subduction channel with kinematic boundary conditions. And 2) an extended model where the overriding plate is included, the channel boundary is deformable and its deflection is not a boundary condition but a model prediction. In my statement that: "One cannot make predictions of the magnitude of the applied stresses in regions outside the model domain" was used in the context of predicting the stress state of the overriding plate. Naturally, one cannot predict the stress state in the first-type of models, as the overriding plate is outside of the model domain. In the second case the stress state of the overriding plate is predicted. The overriding plate IS within the model domain. Therefore, my aforementioned quote is rather stating the obvious and is not a topic for discussion.

A related topic for discussion is the statement by Prof. Platt regarding the model of Marques et al. (2018): "In their model set up, the only load acting on the upper boundary is the weight of the overlying rock". This statement is unrelated to both types of models that Marques et al. (2018) considered. The first type of model considers kinematic boundary condition for the top channel boundary and the second type considers that "[the] top boundary was also left unconstrained, allowing the material to extrude upward freely" p. 1068 in Marques et al. (2018). With respect to Model 1, kinematic boundary conditions at the top channel boundary imply zero velocities on that boundary. Specification of any other loads on that boundary would not be admissible in the

model set up. Stresses on that boundary could be predicted and as such would be an outcome of the model and not an input to the model set up. With respect to Model 2, the top channel boundary is not a boundary of the model domain, and therefore, one cannot specify its load as a boundary condition. Consequently, the loads on the top-channel boundary will be a model prediction and not part of the model configuration. To conclude, none of the two model configurations considered by Marques et al. (2018) has the weight of the overlying rock as the sole load on the top-channel boundary.

---

## Author Comment (AC5) · 30 Nov 2018

Reply to comment by E. Moulas J. P. Platt

In his first paragraph Moulas states "The 50km deflection calculated by Prof. Platt could be hypothetically observed as the result of unloading to a state with negligible tectonic overpressure".

This is incorrect. The elastic deflection of the upper plate results in a restoring force related to the bending moments in the deflected plate. This restoring force increases with the amount of deflection, and the deflection I calculated is such that the restoring

force is sufficient to balance the dynamic pressure in the channel. I proposed this because it is the only physically possible way to achieve force balance across the upper boundary of the channel. The fact that M18 did not include it in their model is the reason I criticized their boundary condition as unphysical. The reason the deflection is so large is because the dynamic pressure proposed by M18 is unrealistic. The rest of Moulas' first paragraph makes no sense.

In his second paragraph Moulas states "Naturally, one cannot predict the stress state in the first-type of models, as the overriding plate is outside of the model domain." This is precisely the problem with the approach taken by M18. Calculating flow and dynamic pressure in a channel that is isolated from its surroundings is completely pointless. The magnitude of the dynamic pressure in the channel is limited by the strength of the upper plate. The maximum possible value is determined by the flexural response, as this assumes the upper plate is strong enough to resist permanent deformation. Any flexural upwarp is unlikely to exceed a few kilometers at most, as otherwise it would have been detected by now from its topographic and gravity signature. A full calculation is needed to determine what magnitude and spatial extent of dynamic pressure is consistent with this limitation, but it is likely to be substantially less than 1.5 GPa. The calculations by Schmalholz in his comment are helpful in this regard. In actuality, however, the response of the upper plate is much more likely to involve permanent viscous and brittle deformation. As I pointed out in my response to Moulas' earlier review, several lines of evidence suggest that the upper plate is unlikely to be able to sustain differential stress in excess of 200 MPa, and this provides the most realistic limitation on the magnitude of the dynamic pressure.

Moulas' third paragraph is based entirely on the misconception discussed above. A model is supposed to be a simplified representation of the real world, allowing calculations that approximate the more complex response of the real system being studied. The model should be consistent with all physical laws, and produce results that can be tested against measurements on the real system. The model set-up by M18 does not

conform with these important principles. In the real world, the load on the upper boundary of the subduction channel will be lithostatic, unless some other physical mechanism is present that can increase that load. Neither M18 nor Moulas have suggested any such mechanism.

With respect to the deformable walls model (model 2), M18 unfortunately did not describe the boundary conditions at all clearly, but careful examination of their Figure 8 indicates that the "unconstrained top boundary" refers to the mouth of the channel, and that the actual upper boundary of the model is fixed, as in model 1.

---

## Author Response (AR1)

**Comment on Marques et al. (2018), Channel flow, tectonic overpressure, and exhumation of high-pressure rocks in the Greater Himalayas, by John P. Platt.**

**Response to Referees.**

5    **Referee 1 (Schmalholz). First Review.**

*Comment*: Platt (2018) questions the main results of Marques et al. (2018), referred to as M2018 in the following, who use a two-dimensional numerical linear-viscous flow model to quantify magnitudes of dynamic pressure in a trapezoidal model domain. A main comment of Platt (2018) is: "I suggest that their estimates of dynamic pressure are at least one order of magnitude too high". This statement, like essentially all other statements in Platt (2018), is purely speculative and not

10  substantiated by any mechanical calculation or alternative numerical model.

*Response*: I have added an analysis of the flexural response of the upper plate to an unbalanced load of 1.5 GPa, which predicts a 50 km flexural upwarp; at least an order of magnitude larger than anything that is observed. I have also added discussion of the likely strength of the upper plate, which is < 120 MPa shear stress. This also limits the ability of the upper plate to confine the dynamic pressure.

15

*Comment*: In point 1 Platt (2018) states: "Whatever the details of the channel geometry, it must ultimately always taper downwards if it is to produce the corner-flow effect". M2018 show with their numerical simulations that return flow is generated in their model, which has an "upward-tapering" geometry. The results of M2018, hence, falsify the above statement of Platt (2018). M2018 never use the term "corner flow", but speak of "channel flow". Corner flow models

20  commonly consider flow around a single corner. With respect to geometry, the trapezoidal model geometry of M2018 is more similar to a circulating cell model (e.g. Pollard and Fletcher, 2005; their figure 10.24).

*Response*: I have deleted my criticism of the description of the channel in terms of an "upward-tapering" geometry. I discuss the use of the term "corner flow", adding a simple description of the physics of this process, and justifying my use of this term to describe the model of M2018.

25

*Comment*: In point 2 Platt (2018) states that the horizontal base of the model in M2018 does not move horizontally which does not fit the tectonic situation in which the Indian lower-crust and mantle lithosphere is underthrusting Tibet and hence the "base" of the Greater Himalayan Sequence should have a horizontal velocity component. This is a fair comment. However, Platt (2018) does not make any prediction about how a horizontally moving base would affect the results of

30  M2018. In the model of M2018 there is a velocity singularity at the left edge of the model base and a model with a horizontally moving base can have a velocity singularity at the right edge of the base. The consequences of such different boundary condition have to be calculated with a corresponding numerical simulation in order to quantify the impact on the results of M2018.

*Response*: I have added a more detailed discussion of the geometrical and kinematic configuration described by M2018, and how it differs from the configuration that they actually use, with the help of a figure. I hope that my revised discussion adds clarity to this important but potentially confusing issue. I don't think it is appropriate that I should be asked to carry out a full numerical simulation to justify this discussion, but in the revised version I point out that simply by inspection it is easy to see that the two configurations will produce dynamic pressures that are of opposite sign.

*Comment*: Point 3: The statement of Platt (2018) that M2018 "do not allow for any motion normal to the channel boundaries" is, to the best of my knowledge, not correct. M2018 also show results for which the material above and below the channel can deform viscously. M2018 state: "This model allows for both channel walls to deform viscously, thus raising the question of how much overpressure they can retain inside the channel". Based on the description of the boundary conditions in section 3.4 of M2018, I conclude that this model allows for motion normal to the upper channel boundary.

*Response*: The section in M2018 on deformable walls is unfortunately very difficult to follow, as they do not define the thickness or geometry of the walls, and their description of the boundary conditions is confusing and ambiguous. It appears that they have incorporated a layer of relatively high viscosity material into the model, above and below the channel. The model as a whole still has fixed upper and lower boundaries, however, so the system behaves in much the same way as the model without the deformable walls, and the predicted dynamic pressure is almost identical. I discuss this point in the revised version.

*Comment*: The paragraph on page 2 from lines 6 to 18 in Platt (2018) includes mainly speculative "should-would-could" arguments, which are also mechanically unsound. For example, Platt (2018) argues that "an unbalanced upward load of 1.5 GPa should cause a substantial flexural upwarp of the upper plate, possibly tens of km in amplitude, producing a major topographic and gravity anomaly". It is not logical why there should be an "unbalanced upward load" in a mechanical model, which is based on the equations of force balance. The dynamic pressure of 1.5 GPa is not an "unbalanced load"; this dynamic pressure and the associated pressure gradient is responsible for "pushing" the viscous material upwards, against gravity and against the downward direction of the applied boundary velocity.

*Response*: This comment addresses my fundamental disagreement with M2018 and with the referees. Fixed boundaries are commonly assumed in fluid mechanics problems, because the mechanical contrast between a low-viscosity fluid such as water and a steel pipe, for example, is so large that deformation of the boundaries can be neglected. In the case of the subduction channel modeled by M2018 in their Figure 2, the viscosity is 24 orders of magnitude greater than that of water, and the viscous stresses are correspondingly larger. If a dynamic pressure of 1.5 GPa is applied from below to the upper boundary of the channel, a physical mechanism is required that is capable of keeping the boundary fixed, and M2018 give no indication what this might be. I argue that the only mechanism that can balance forces across the boundary is the flexural response of the upper plate. I have added a section to the revised version that explains this point in detail.

*Comment*: Platt (2018) further argues that "given that the material in the subduction channel is incompressible, even a small amount of flexural displacement would be enough to relieve the dynamic pressure". Indeed, it is well established that the dynamic pressure depends on the strength of the channel walls and dynamic pressure decreases when channel walls get weaker and, hence, displace more (e.g. Mancktelow, 2008). Such pressure relieve has been quantified with numerical models in several studies (e.g. Mancktelow, 2008) and is also mentioned in M2018 in section 1.2. M2018 report significant dynamic pressure also for models in which the viscosity of the channel was 100 to 1000 times smaller than the viscosity of the material bounding the channel (their section 3.4 and their figure 8). Therefore, the elastic flexural displacement, mentioned by Platt (2018), has to be calculated with an adequate model in order to test whether and for what conditions elastic flexure causes a significant pressure relieve.

*Response*: I have deleted from my comment the discussion of the possible response of the channel to deformation of the hangingwall.

*Comment*: Page 2, lines 23-24. The statement of Platt (2018), "I suggest that their estimates of dynamic pressure are at least one order of magnitude too high", is not substantiated and not quantified by a mechanical calculation or model. I recommend to calculate dynamic pressure and not to suggest it.

*Response*: See my response to the referee's first comment.

*Minor comment*s. I have deleted all the sections of my Comment concerning petrological issues, except to point out that the model of M2018 does not provide an adequate basis to disccuss Himalayan metamorphism.

**Referee 1 (Schmalholz). Second Review.**

*Comment*: M2018 use a model with kinematic boundary conditions in which they set the velocities of the upper channel wall to zero. Platt argues that this boundary condition is non-physical. Actually, most corner and channel flow models applied in the Earth Sciences (e.g. flow in a mantle wedge, flow mélanges or return flow in a subduction channel; e.g. Cloos, 1982; England and Holland, 1979; Turcotte and Schubert, 2014) are based on such kinematic boundary condition. Usually, a corner/channel-parallel velocity is applied on one side and on the other side, typically the hanging wall, the velocities are zero. This boundary condition of zero velocity and the mechanical constraint of force balance require that the loads normal to the upper channel boundary, which are exerted by the channel, must be equal to the loads normal to the upper channel boundary exerted by the hanging wall. However, Platt argues that "The loads normal to the upper boundary of the channel consist of the pressure in the channel (lithostatic load + dynamic pressure) on one side, and the lithostatic load alone on the other side". This statement of Platt is wrong because it violates the force balance across the upper channel wall. Force balance requires that the load (more precisely the total stress normal to the boundary) of the hanging wall must balance the load of the channel (for slow deformation if inertial forces are neglected). It is, hence, not the boundary condition of M2018, which is non-physical, it is the assumption of Platt of a lithostatic load in the hanging wall which is non-physical in the

context of the model of M2018. Therefore, I argue that Platt's argument is mechanically not sound, because the stress state he assumes violates the force balance across the upper channel wall.

*Response*: It is actually a bit odd to say that my argument is unsound because it violates force balance, when my argument was that the boundary condition in M2018 violates force balance! As noted above, if a dynamic pressure of 1.5 GPa is
5    applied from below to the upper boundary of the channel, a physical mechanism is required that is capable of keeping the boundary fixed, and M2018 give no indication what this might be. I argue that the only mechanism that can balance forces of this magnitude across the boundary is the flexural response of the upper plate, but for this to produce a force that could balance the dynamic pressure, there has to be a finite deflection. The only other possible response is an upward acceleration of the upper plate, which is inconsistent with it being fixed. Hence my argument that the boundary condition is unphysical.
10    I have tried to clarify these points in my revised Comment.

*Comment*: Schmalholz has added an extensive discussion of the flexural response of the upper plate, and has proposed situations that could reduce the amount of flexure to reasonable values. He concludes "Therefore, I argue that the strong statement of Platt, that estimates of dynamic pressure of M2018 are at least one order of magnitude too high, is not justified."
15    *Response*: This section is an implicit recognition that my suggestion of a flexural response to the dynamic pressure was neither speculative nor mechanically unsound, as Schmalholz indicated in his initial review. I agree that the analysis is sensitive to the values taken for $L$ and $h$, but I note that the value for $L$ I used was measured directly from Figure 2 of M18, being the distance along the upper boundary of the channel over which the dynamic pressure exceeds 1.5 GPa, and my value for $h$ was taken from Jordan & Watts (2005), who determined the effective elastic thickness over the whole region from
20    topography and Bouguer gravity data. 20 km is the maximum value for southern Tibet: they give the range as 0-20 km. To get a 60 km value for $h$ we would need to go into the Indian shield, which is composed of dry granulite facies rocks overlying cold lithospheric mantle. I agree that the precise amount of the elastic deflection is open to discussion, but using the values I determined for $L$ and $h$, the dynamic pressure would have to be reduced to 60 MPa to get a flexural deflection of 2 km, which might be small enough to escape detection. I elaborate on this point in my revised Comment.

25

*Comment*: The next section of Schmalholz's comment starting "The elastic beam model is not very suitable to quantify the mechanical resistance of the hanging wall" is mainly concerned with possible modifications to the geometry of the model of M2018 that could reduce the flexural response to reasonable levels. It then continues to consider processes in shear zones and at the scale of individual mineral grains that could produce some sort of overpressure.
30    *Response*: I agree with some of these points, and disagree with others, but this does not affect my Comment, which is concerned with the model presented by M2018. I would note that M2018 do not even consider a flexural response, so we cannot use that possibility to justify their boundary condition, which assumes that the upper boundary is fixed.

*Comment:* Concerning point 3 of my review whether the model of M2018 with a viscously deforming hanging wall allows motion normal to the upper channel boundary: The authors of M2018 have already replied to Platt and they confirmed my evaluation that the comment of Platt, claiming that motion normal to the channel wall was not possible, is incorrect. I trust that the authors of M2018 have checked their model and boundary condition before their reply to Platt.

5 *Response*: As pointed out above, this section of M2018 is very confusing and ambiguous. My assessment is that the authors of M2018 have simply dismissed my criticisms without evaluating them, and that their deformable walls model has the same non-physical boundary condition as the others. I have discussed this point as clearly as I can in the revised version.

**Referee 2 (Moulas). First review.**

10 *Comment:* The comment posted by Prof. Platt (hereafter P18) highlights some points of the model proposed by Marques and co-workers (hereafter M18) in their publication entitled "Channel flow, tectonic overpressure, and exhumation of high-pressure rocks in the Greater Himalayas" in a very critical manner. To the author's opinion, the most essential criticism of P18 on M18 model is the model configuration. Based on P18, the contact between the subduction channel and its overriding plate, as presented in the model of M18, is "unphysical" and it leads to erroneous predictions of tectonic overpressure (TOP).

15 The characterization "unphysical" and the subsequent arguments used by P18 are based on erroneous assumptions that lead to unphysical conclusions, and are therefore unjustified.

*Response*: I have addressed these issues in my responses to Schmalholz's comments, including the points raised by Moulas about the model with deformable walls.

20 *Comment:* Following P18, the deflection of the overriding plate is a consequence of having "unbalanced loads" in the channel. This criticism by P18 reveals a misconception of P18 regarding force balance in Stokes' equations. The model of M18 satisfies force balance everywhere within the model.

*Response*: My criticism was that the forces are unbalanced across the upper boundary, not within the channel.

25 *Comment:* One cannot make predictions of the magnitude of the applied stresses in regions outside the model domain.

*Response*: M2018 presented their model as a calculation of the dynamic pressure in a real subduction channel in the Himalayas, and they draw conclusions from it about Himalayan metamorphism. They cannot restrict their analysis to the subduction channel and pretend that the rest of the universe doesn't exist. We have to ask whether the upper boundary condition for their model is consistent with what we know about the tectonic setting. I make the case that it is not.

30

*Comment:* Platt argued that the TOP in the viscous channel is unbalanced since the overriding plate experiences lithostatic pressure. The last statement clearly reveals a mechanical misconception, i.e. it was the implicit assumption of P18 that pressure is lithostatic in the overriding plate. The last statement is mechanically unfeasible and violates force balance. In other words, there is no moment in time where there would have been significant TOP in the channel and lithostatic stress in

the channel wall. Therefore, if there is a significant TOP in the channel then one needs to solve for the state of stress outside of the channel in order to have any meaningful stress estimates. In summary, the large values of TOP predicted by M18 are just the outcome of model inputs regarding the geometry, the specific rheology, the overall boundary conditions etc. How appropriate are these estimates needs to be verified and quantified. Without specific information on the stress distribution on
5 the models of M18, one cannot judge how realistic these results are with respect to their stress magnitudes and the strength that they imply for the overriding plate.

*Response*: In the absence of any tectonic mechanism to increase stress in the upper plate, it is reasonable to assume that the pressure is lithostatic. M2018 make no suggestion about this, and neither does Moulas. Moulas' comment that the situation is mechanically unfeasible and violates force balance is precisely my criticism of M2018. My suggestion of a flexural
10 response of the upper plate to the applied load was an attempt to find a solution, and to discover whether their estimates of dynamic pressure are appropriate. I conclude that they are not. I would also argue that it is the responsibility of the authors of M2018 to find out how realistic their results are and what the implications are for the strength of the upper plate. In the face of considerable resistance, I have been attempting to do this for them.

15 *Specific comments.*

P1-l.15-25, M18 did not state that they have a typical corner-flow model. Therefore, there is no justification for the suggestions of P18 regarding the tapering angles of the models of M18.

*Response*: It is a corner flow model. The channel has to close downward, and M18 state this. I have removed my comments on the taper from my revised comment.
20

P1-l.28. There is no specific reason on why the downward velocity (of the footwall ramp) must be exactly the same as the one of the plate especially when rheological boundaries are considered. Perhaps, having it perfectly immobile like in the case of M18 may be an exaggeration.

*Response*: M18 state that the underthrust plate is rigid. The footwall ramp is part of the underthrust plate, so it must move
25 with it. M18 make no reference to "rheological boundaries". They take a feature that they describe as part of the lower plate, and give it the same velocity as the upper plate. That isn't an exaggeration: it's simply wrong.

P2-l.3-5 In their paper, M18 clearly state that they consider also the case of deformable walls therefore all this section is not justified.

30 P2.l.6-24 All this part is speculative and based on erroneous implicit assumptions. i.e. there is no reason why the load must be unbalanced. Therefore, all the arguments that follow (e.g. about unrealistically large flexures etc.) are based on faulty assumptions.

*Responses*: I have addressed both these issues in my responses to Schmalholz's comments.

With respect to Moulas' remaining comments, I have removed the sections in question from the revised version.

**Referee 2 (Moulas). Second review.**

Much of this review is so confused that I have difficulty in formulating responses. I have extracted sentences that I
understand, and responded to them as best I can.

*Comment:* The 50km deflection calculated by Prof. Platt could be hypothetically observed as the result of unloading to a state with negligible tectonic overpressure (TOP).

*Response:* This is incorrect. The elastic deflection of the upper plate results in a restoring force related to the bending moments in the deflected plate. This restoring force increases with the amount of deflection, and the deflection I calculated is such that the restoring force is sufficient to balance the dynamic pressure in the channel. I proposed this because it is the only physically possible way to achieve force balance across the upper boundary of the channel. The fact that M18 did not include it in their model is the reason I criticized their boundary condition as unphysical. The reason the deflection is so large is because the dynamic pressure proposed by M18 is unrealistic.

*Comment:* Naturally, one cannot predict the stress state in the first-type of models, as the overriding plate is outside of the model domain.

*Response:* This is precisely the problem with the approach taken by M18. Calculating flow and dynamic pressure in a channel that is isolated from its surroundings is completely pointless. The magnitude of the dynamic pressure in the channel is limited by the strength of the upper plate. The maximum possible value is determined by the flexural response, as this assumes the upper plate is strong enough to resist permanent deformation. Any flexural upwarp is unlikely to exceed a few kilometers at most, as otherwise it would have been detected by now from its topographic and gravity signature. A full calculation is needed to determine what magnitude and spatial extent of dynamic pressure is consistent with this limitation, but it is likely to be substantially less than 1.5 GPa.

*Comment:* Kinematic boundary conditions at the top channel boundary imply zero velocities on that boundary. Specification of any other loads on that boundary would not be admissible in the model set up. Stresses on that boundary could be predicted and as such would be an outcome of the model and not an input to the model set up.

*Response:* This is an attempt to defend the indefensible. If you set up a model that predicts a 1.5 GPa pressure in the crust in excess of lithostatic, without having a physical mechanism to confine that pressure, then it's a bad model.

All these points are addressed in the revised version of my Comment.

**Revised version with markup:**

[revised manuscript text omitted]

Schmalholz, Medvedev, S., Lechmann, S., and Podladchikov, Y.: Geophysical Journal International, 197, 680–696, 2014.

Spiegelman, M., and McKenzie, D.: Simple 2-D models for melt extraction at mid-ocean ridges and island arcs, Earth and Planetary Science Letters, 83, 137-152, 1987.

Turcotte D. L, Schubert, G.: Geodynamics, 2nd ed., Cambridge University Press, 456pp, 2002.

Watts, A. B., and Zhong, S.: Observations of flexure and the rheology of oceanic lithosphere, Geophysical Journal International, 142, 855-875, 2000.

**Figure Caption**

Figure 1.  A) Downward tapering subduction channel illustrating the configuration that can lead to corner flow and positive dynamic overpressure ($\Delta P$). B) Geometrical and kinematic configuration of the Himalayan subduction zone as described by Marques et al. (2018). The base of the channel moves with the lower plate, and $\Delta P$ is negative.  C) Configuration used for calculations in the model by Marques et al. (2018).  The base is attached to the upper plate.

---

## Referee Report (RR1)

Review of "Comment on Marques et al. (2018), Channel flow, tectonic overpressure, and exhumation of high-pressure rocks in the Greater Himalayas" by John Platt.

Dear editor,

please find below my specific comments regarding the criticism of Prof. Platt on the paper by (Marques et al., 2018). The author has revised some major points that were raised by my previous review, however I have some major points to add which I list in detail below. Most importantly, I would like to highlight that Prof. Platt uses an analytical solution of elastic deformation in order to criticize the purely viscous model proposed by Marques et al (2018). Clearly, different assumptions are lying behind the different models (elastic vs. viscous) and therefore I recommend that the author separates what is actually the result of Marques et al (2018) analysis and what is a consequence of his own model. In particular, Prof. Platt uses an elastic model to predict the flexure that would result from the stresses calculated by the viscous model of Marques et al. (2018). At this point, I would like to highlight (see also my point P8) that the geometrical configuration used by Marques et al. (2018) approximates the one currently observed, and not an initial condition. Consequently, in order to predict the time evolution of such a system and check out how realistic it is, one can model the time evolution of the system. Any change in geometry and viscosity structure of a purely viscous system will result to different stress/velocity distribution.

Best

Evangelos Moulas

Specific Comments (I refer to P1, P2 as points 1, 2 etc)

P.1 "Note that dynamic overpressure as used here is generated by flow in a viscous fluid, and differs in this respect from the more widely recognized concept of tectonic overpressure, which is related directly to deviatoric stress, and can exist in a static situation, with or without deformation".
>> This statement is confusing. The flow of a viscous fluid cannot be unrelated to the deviatoric stresses. By definition, the flow of viscous materials requires the presence of deviatoric stresses.

P2. "Return flow in subduction channels has been proposed as a mechanism for exhuming high-pressure metamorphic rocks from deep in the subduction zone (e.g., Cloos, 1982). Possible drivers are buoyancy (e.g., England & Holland, 1979; Beaumont et al., 2009), topographic gradients (e.g., Beaumont et al., 2001), or dynamic overpressure (e.g., Gerya & Stockhert, 2002)."
>> In the case of corner flow (Cloos, 1982), the return flow is independent of buoyancy stresses (Batchelor, 1967). In fact, the dynamic overpressure (difference from the lithostatic; also associated with deviatoric stresses) is responsible for the return flow. In other words, there is no corner flow s.s. without pressure deviations from the lithostatic. By contrast, the main driver for exhumation in the channel-flow model of England and Holland (England and Holland, 1979), is buoyancy. I would therefore recommend rephrasing of the related paragraph.

P3. "the dynamic overpressure is limited by the ability of the channel walls to contain it. If the walls deform, the pattern of flow will change, and the dynamic overpressure is likely to decrease."

>> The author has a point here, however one needs to model the time evolution of the wall deformation. For example, a system where the wall deflects in 10,000 years is different from a system where the wall deformation would take tens of millions of years to evolve. The specifics of the evolution would, in turn, depend on the particular mechanical response of the wall and the boundary conditions assumed. Therefore, without being more specific this point is rather weak.

P4. "The second problem is that they assume a fixed upper boundary to the subduction channel, which cannot be defended in geological terms, and leads to unrealistic conclusions"

>> I have stated my disagreement with this comment in my previous review. The author in one of his response comments suggested that a careful investigation of the boundary conditions of Marques et al (2018) reveals that the wall is fixed. Based on the description of the model setup with deformable walls by Marques and co-workers, I find this statement misleading (i.e. in the models with deformable walls the walls are not fixed).

P5. "A more fundamental problem concerns their use of a fixed upper boundary to the channel. It is true that fixed boundaries are commonly assumed in fluid mechanics problems, because the mechanical contrast between a low-viscosity fluid such as water and a steel pipe, for example, is so large that deformation of the boundaries can be neglected. In the case of the subduction channel modelled by M2018 in their Figure 2, the viscosity is orders of magnitude greater than that of water, and the viscous stresses are correspondingly larger."

>> Fluid dynamics solutions are not restricted to water; in fact, it is pointless to use water as a reference. This is actually why fluid mechanics are successfully applied to structural geology and geodynamics problems (Pollard and Fletcher, 2005; Turcotte and Schubert, 2014). Fluid dynamics solutions depend on the viscosity ratios of different materials. Even when a rock has a viscosity much larger than that of water, it can still behave as a low-viscosity fluid compared to the rock that exhibits even higher viscosity (see for example Gerya, 2010, p. 245). Marques and co-workers used viscosity ratios differing by 2-3 orders of magnitude. When the viscosity of the wall is 3 orders of magnitude or larger than the viscosity of the convecting fluid, then, the deformation of the wall would be negligible. Importantly, even if the initial boundary is assumed perfectly straight, time integration of the mechanical solution allows for conclusions to be drawn on the deformation of the strong lid.

P6 "If a dynamic overpressure of 1.5 GPa is applied from below to the upper boundary of the channel, a physical mechanism is required that is capable of keeping the boundary fixed, and M2018 give no indication what this might be."
>> This statement is not true. Marques and co-workers clearly state that this can occur if the walls are strong, so that the boundary would behave as if the lid were rigid. It is the high viscosity of the channel wall (that can build up large stresses) that is responsible for keeping the boundary nearly fixed.

P7 "the only load acting downwards on the upper boundary is the lithostatic pressure"
>> This statement cannot be true for a deforming lithosphere with topography and density changes.

P8 "In the case of a subduction channel, the configuration can be approximated by the analysis for flexural doming above an igneous intrusion presented by Turcotte & Schubert (2002)."
>> The applicability of this solution in the subduction channel is not entirely justified. An important assumption for the application of the solution of Turcotte and Schubert (2002) is that the initial condition is known. Firstly, the layers of rocks are assumed to be horizontal and secondly, the deflections are calculated from this initial stage. By contrast, the configuration of Marques and co-workers is not an initial condition. Their solution is meant to depict the current configuration that satisfies force balance. Therefore, the plate deflection evolution in the Marques et al (2018) viscous model must be integrated over time as it is commonly done in Geodynamic modelling of slow viscous flow e.g. Gerya (2010).

P9. "Various lines of evidence suggest than an upper limit of ~120 MPa shear stress is reasonable for continental lithosphere in actively deforming regions"
>> These values for shear stress have no universal applicability. There are numerous models that use experimentally determined flow laws that would not agree with such a statement. Stresses on the order of 100MPa are the minimum required to support topography in mountainous regions only if the entire lithosphere is stressed in a uniform manner (average stresses). Clearly, this is highly improbable since the presence of viscosity heterogeneities would result to regions in which the shear stress would be significantly higher or lower (Schmalholz et al., 2018, 2014).

P10. "The model set-up by M2018 does not conform with these important principles …."
>> As mentioned in my previous review, instead of comparing the results of the Marques et al. (2018) model with natural observations, Prof. Platt compares the results of his own elastic flexure model with natural observations. However, the assumptions lying behind the elastic flexure formula are different to those invoked by Marques and co-authors (2018) in their viscous-flow model.

References:

Batchelor, G.K., 1967. An Introduction to Fluid Dynamics. Cambridge University Press.
Cloos, M., 1982. Flow melanges: Numerical modeling and geologic constraints on their origin in the Franciscan subduction complex, California. Geological Society of America Bulletin 93, 330–345. https://doi.org/10.1130/0016-7606(1982)93<330:FMNMAG>2.0.CO;2
England, P.C., Holland, T.J.B., 1979. Archimedes and the Tauern eclogites: the role of buoyancy in the preservation of exotic eclogite blocks. Earth and Planetary Science Letters 44, 287–294. https://doi.org/10.1016/0012-821X(79)90177-8
Gerya, T., 2010. Introduction to Numerical Geodynamic Modelling. Cambridge University Press.
Marques, F.O., Mandal, N., Ghosh, S., Ranalli, G., Bose, S., 2018. Channel flow, tectonic overpressure, and exhumation of high-pressure rocks in the Greater Himalayas. Solid Earth 9, 1061–1078. https://doi.org/10.5194/se-9-1061-2018
Pollard, D.D., Fletcher, R.C., 2005. Fundamentals of Structural Geology. Cambridge University Press.
Schmalholz, S.M., Duretz, T., Hetényi, G., Medvedev, S., 2018. Distribution and magnitude of stress due to lateral variation of gravitational potential energy between Indian lowland and Tibetan plateau. Geophysical Journal International ggy463–ggy463. https://doi.org/10.1093/gji/ggy463
Schmalholz, S.M., Medvedev, S., Lechmann, S.M., Podladchikov, Y., 2014. Relationship between tectonic overpressure, deviatoric stress, driving force, isostasy and gravitational potential energy. Geophysical Journal International 197, 680–696. https://doi.org/10.1093/gji/ggu040
Turcotte, D.L., Schubert, G., 2014. Geodynamics, 3rd ed. Cambridge University Press.

---

## Author Response (AR2)

**Comment on Marques et al. (2018), Channel flow, tectonic overpressure, and exhumation of high-pressure rocks in the Greater Himalayas:  Response to Review by the Topical Editor**

*Comment*:  Dear John,

In order to check your arguments about boundary conditions numerically (as suggested by one of the reviewers), I did numerical experiments for a weaker channel surrounded by stronger rocks under condition of free lithospheric surface present in the model with inclined gravity orthogonal to this surface. Rock rheology is viscoelastic. The respective matlab Viscoelastic2D_m code is copied below for your reference. GPa-level overpressure indeed forms in the channel at high channel viscosity ($10^{22}$ Pa*s) that are balanced by GPa-level deviatoric stresses (and under/over pressures) in surrounding stronger rocks. It remains questionable if such stresses can be sustained by these rocks but within the referred visco-elastic approximation the results are rather consistent with the paper of Marques.

Therefore I think that your statement that "The fixed upper boundary condition in their models violates force balance and is unphysical" is not correct. Having this boundary non-fixed does not preclude overpressure (if surrounding rocks are much stronger than the channel).

Therefore, I think your comment cannot be published in its present state.

With my best wishes.

Taras

*Reply*:  I appreciate very much the Topical Editor's code, which I ran, and this confirmed that GPa-level overpressure forms in the channel even with a free upper surface, using a sufficiently high viscosity for the channel and the upper plate.  I note, however, that the upper boundary of the channel does not remain fixed, as in the model by M2018; it flexes upwards, and so does the free upper surface.  As the Editor points out, large values of tectonic overpressure are developed in the upper plate, reflecting its relatively high strength, which enables it to confine the dynamic overpressure in the channel.

I have therefore removed any mention of non-physicality, and revised the manuscript to make the point that the upper boundary condition cannot be reconciled with what we know about the mechanical properties of the Himalayan upper plate.

I hope that the revised version will now be acceptable for publication.

[revised manuscript text omitted]

<table>
<tr><td>**Deleted:** A fixed upper boundary is therefore non-physical</td></tr>
</table>

<table>
<tr><td>**Deleted:** A load of 1.5 GPa is likely to cause permanent deformation in the hanging wall; in the absence of such deformation, the upper plate should flex elastically.</td></tr>
</table>

$$D = \frac{Eh^3}{12(1-v)},$$ where $E$ is Young's modulus, $h$ is the effective elastic thickness of the upper plate, and v is Poisson's ratio. I

estimate the following values, based on Figure 2A from M2018, for the region between 40 and 100 km depth in the subduction zone:

$L = 175$ km;

$p = 1.5$ GPa averaged over $L$. For the mechanical parameters, I have taken the following values from Jordan and Watts (2005) for the upper plate:

$E = 10^{11}$ Pa,

$h = 20$ km (Jordan and Watts give a range from 0 – 20 km for the effective elastic thickness in southern Tibet, so I have taken a conservative value),

$v = 0.25$.

The predicted deflection is 50 km: this is what is required to produce a restoring force equal to the upward load of 1.5 GPa predicted by M2018. The deflection is so large that it violates one of the assumptions of the analysis, that $w$ is small compared to $L$. The analysis does not take into account the tapering geometry of the upper plate (which will increase the deflection), and it is sensitive to the values chosen for $E$ and $h$. But it is sufficient to demonstrate that a dynamic overpressure of 1.5 GPa in the Himalayan subduction zone is geologically unsustainable. No flexural upwarp of ~50 km amplitude has been detected in southern Tibet. To achieve a more reasonable value for the deflection (say 2 km) we would need either to choose a value of 60 km for $h$, or to reduce the dynamic overpressure to <60 MPa. A value of 60 km for the effective elastic thickness is characteristic of the Indian plate, which is made up of granulite facies crustal rocks overlying thick and cold lithospheric mantle, but it is quite outside the range of values found for Tibet and the upper plate of the

Himalayas.

*Deformable walls*

In practice, the rocks in the upper plate of the Himalayas are more likely to deform permanently if subjected to significant dynamic overpressure. M2018 recognize that some permanent deformation is likely, and they attempt to address this with their deformable walls model. This section of their paper is very difficult to follow, as they do not define the thickness or geometry of the walls, and their description of the boundary conditions is confusing and ambiguous. It appears that they have incorporated a layer of relatively high viscosity material into the model, above and below the channel. The model as a whole still has fixed upper and lower boundaries, however, so the system behaves in much the same way as the model without the deformable walls, and the predicted dynamic overpressure is almost identical. The problem with boundary conditions discussed in the previous section remains unchanged, and little of value can be inferred from this model.

*Permanent deformation in the upper plate*

| Deleted: flexural |

| Deleted: Lower values for the flexural upwarp could be obtained with lower values and spatial extents of the dynamic overpressure, as shown by Schmalholz in the Discussion session. The values I have used were taken directly from M2018. |

[revised manuscript text omitted]

---

## Author Response (AR3)

**Platt Discussion:  Referee Responses**

**Responses to Referee 1.**

I find the revised version of the comment by Platt now acceptable for publication. After the online discussions (which are in my opinion useful and interesting) related to the first version of the comment of Platt I am looking forward to the reply of Marques and co-authors. I have only a few comments to the revised comment of Platt.

Line 22-23, page 3: I find this sentence misleading. If the resistance of the plate becomes larger, then the corresponding deflection, due to a given normal load, becomes smaller. Therefore, an extremely small deflection can correspond to an extremely large resistance. The boundary condition of a straight channel wall with zero deflection applied by Marques et al. (2018) can be justified by two assumptions: (i) the deflection is small with respect to the length of the channel wall so that it is negligible. (ii) the straight channel wall corresponds to the elastically-deflected state implying that without dynamic pressure the upper channel wall would be curved.

Agreed.  The confusion was caused by my use of the word "resistance", which I was using to refer to the restoring forces created by the bending moments produced by the deflection.  I have deleted the sentence and modified the text.
Note that the calculated deflection is not small (50 km over an up-dip length of 175 km), and it is improbable that the channel wall could have had that degree of curvature initially.

Line 2-11, page 5: This is a selective view on the topic of the strength of the lithosphere. There are many studies suggesting much higher shear stresses in a deforming continental lithosphere, especially when flexure is considered. Many studies on the strength of the lithosphere considering so-called yield-strength envelopes, based on flow laws from rock deformation experiments, exhibit one or more levels in the lithosphere with stresses corresponding to shear stresses significantly larger than 120 MPa (e.g. Burov, 2011, and references therein). For example, if strength in the hanging wall in depths larger than 40 km would be dominated by diabase then the stress could be significantly larger than 120 MPa (e.g. Burov, 2011, his figure 7). Furthermore, flow laws for dry anorthite indicate that stresses in the lower crust could be significantly larger than 120 MPa (e.g. Rybacki et al., 2006).

I agree that there are different views on the strength of the lithosphere, which largely reflect the fact that the lithosphere varies a lot in both composition and thermal structure.  An intermediate to mafic  lower crust dominated by feldspar, pyroxene, and garnet, with a low thermal gradient, can probably support >800 MPa differential stress near the Moho (e.g., Platt & Behr, GRL, 2011, Figure 2).  But there is no evidence that the upper plate in the Himalayas had this composition. The velocity structure reported from the INDEPTH profile across southern Tibet suggests that the composition is typical of mid-crustal rocks (e.g., granite and metamorphic rocks with hydrous mineral assemblages) (Haines et al., Tectonics, 2003).  As I point out in the text, at depths of 50 km or more in the Himalayas, these rocks were close to their melting point, and would have been very weak.  I have added a few sentences to the text on this point.

Line 14-16, page 5: This suggestion that "the values of dynamic overpressure calculated by

M2018 are at least an order of magnitude too high" is based on the assumption that the rocks forming the hanging wall exhibit shear stresses smaller than ca. 120 MPa. Marques et al. (2018) are aware of the fact that the magnitude of the dynamic pressure depends on the effective strength of the hanging wall, and they write in their manuscript: "Ultimately, tectonic overpressure (TOP) can only exist if the channel walls are strong enough.". The effective strength of the hanging wall is a key parameter controlling the magnitude of dynamic pressure in the model of Marques et al. (2018). I agree that a dynamic pressure of 1.5 GPa within large channel regions, as suggested by Marques et al. (2018), is very difficult to justify. However, a magnitude of the dynamic pressure of 0.3 to 0.5 GPa is a factor of 5 to 3 times smaller than the value of 1.5 GPa and I think that values between 0.3 and 0.5 GPa for dynamic pressure cannot be excluded based on currently available flow laws for the lower crust. Hence, I still think that the above "suggestion" that values of dynamic overpressure are "at least an order of magnitude too high" is also very difficult to justify.

The statement that "The effective strength of the hanging wall is a key parameter controlling the magnitude of dynamic pressure in the model of Marques et al. (2018)" is a bit too generous. By assuming a rigid upper boundary they effectively assign an infinite strength to the upper plate. Their model with "deformable walls" is misleading:  what they have done is to incorporate higher viscosity layers on either side of the channel into the model domain, but they retain the fixed boundary conditions above and below the model domain. This model therefore fails to test the effect of limited strength in the hangingwall as a whole.  Taras Gerya kindly provided me with a Matlab code that models the system described by M2018, but using a free upper boundary to the upper plate, instead of a fixed boundary to the channel. In this model the upper boundary of the channel is pushed upwards by the overpressure in the channel, creating a substantial topographic deflection of the free surface.  The magnitude and dimensions of the region of dynamic overpressure is greatly reduced as a result.
As discussed above, I would defend a strength limit of ~120 MPa for the upper plate, but I have softened the statement about the overpressure.

**Responses to Referee 2.**

Prof. Platt uses an analytical solution of elastic deformation in order to criticize the purely viscous model proposed by Marques et al (2018).

I didn't criticize the viscous model as such, I criticized the use of a rigid boundary condition. For a model to have any applicability to the real world, the boundary conditions must correspond at least approximately to the constraints that the real world would impose.  The assumption of a rigid upper boundary is equivalent to a statement that the upper plate of the Himalaya is infinitely strong.  My point is that even if we neglect permanent deformation, the upper plate will respond elastically, and we have enough information to calculate that response.

I would like to highlight (see also my point P8) that the geometrical configuration used by Marques et al. (2018) approximates the one currently observed, and not an initial condition.

I discuss this issue under point P8

P.1  "Note that dynamic overpressure as used here is generated by flow in a viscous fluid, and differs in this respect from the more widely recognized concept of tectonic overpressure, which is related directly to deviatoric stress, and can exist in a static situation, with or without deformation". This statement is confusing. The flow of a viscous fluid cannot be unrelated to the deviatoric stresses. By definition, the flow of viscous materials requires the presence of deviatoric stresses.

Yes, the flow of a viscous fluid requires deviatoric stress, but the dynamic pressure is not calculated from the deviatoric stresses themselves.  It is calculated using the Navier-Stokes equations, which relate the pressure *gradient* to the deviatoric stress gradients.  This differs from the definition of tectonic overpressure in the usual sense, where one of the principal stresses (usually the minimum) is assumed to be equal to the lithostatic load, and the overpressure is calculated as the difference between the mean stress and the principal stress in question, which by definition is the deviatoric stress in that plane.

P2.  "Return flow in subduction channels has been proposed as a mechanism for exhuming high-pressure metamorphic rocks from deep in the subduction zone (e.g., Cloos, 1982). Possible drivers are buoyancy (e.g., England & Holland, 1979; Beaumont et al., 2009), topographic gradients (e.g., Beaumont et al., 2001), or dynamic overpressure (e.g., Gerya & Stockhert, 2002)." In the case of corner flow (Cloos, 1982), the return flow is independent of buoyancy stresses (Batchelor, 1967). In fact, the dynamic overpressure (difference from the lithostatic; also associated with deviatoric stresses) is responsible for the return flow. In other words, there is no corner flow s.s. without pressure deviations from the lithostatic. By contrast, the main driver for exhumation in the channel-flow model of England and Holland (England and Holland, 1979), is buoyancy. I would therefore recommend rephrasing of the related paragraph.

Agreed – the model of Cloos (1982) does involve dynamic overpressure, and I have modified the text to make that clear.  But return flow, in the more general sense, may commonly involve several drivers, and the buoyancy of a low-density fluid in the subduction channel is one of them.

P3.  "the dynamic overpressure is limited by the ability of the channel walls to contain it. If the walls deform, the pattern of flow will change, and the dynamic overpressure is likely to decrease."   The author has a point here, however one needs to model the time evolution of the wall deformation. For example, a system where the wall deflects in 10,000 years is different from a system where the wall deformation would take tens of millions of years to evolve. The specifics of the evolution would, in turn, depend on the particular mechanical response of the wall and the boundary conditions assumed. Therefore, without being more specific this point is rather weak.

I discuss the specifics later in the text.

P4. "The second problem is that they assume a fixed upper boundary to the subduction channel, which cannot be defended in geological terms, and leads to unrealistic conclusions"
I have stated my disagreement with this comment in my previous review. The author in one of his response comments suggested that a careful investigation of the boundary conditions of

Marques et al (2018) reveals that the wall is fixed. Based on the description of the model setup with deformable walls by Marques and co-workers, I find this statement misleading (i.e. in the models with deformable walls the walls are not fixed).

In most of the models the boundaries are clearly stated to be fixed.  The model with "deformable walls" is misleading:  what M2018 have done is to incorporate higher viscosity layers on either side of the channel into the model domain, but they retain the fixed boundary conditions above and below the model domain. This model therefore fails to test the effect of limited strength in the hangingwall as a whole.

P5. "A more fundamental problem concerns their use of a fixed upper boundary to the channel. It is true that fixed boundaries are commonly assumed in fluid mechanics problems, because the mechanical contrast between a low-viscosity fluid such as water and a steel pipe, for example, is so large that deformation of the boundaries can be neglected. In the case of the subduction channel modelled by M2018 in their Figure 2, the viscosity is orders of magnitude greater than that of water, and the viscous stresses are correspondingly larger."
Fluid dynamics solutions are not restricted to water; in fact, it is pointless to use water as a reference. This is actually why fluid mechanics are successfully applied to structural geology and geodynamics problems (Pollard and Fletcher, 2005; Turcotte and Schubert, 2014). Fluid dynamics solutions depend on the viscosity ratios of different materials. Even when a rock has a viscosity much larger than that of water, it can still behave as a low-viscosity fluid compared to the rock that exhibits even higher viscosity (see for example Gerya, 2010, p. 245). Marques and co-workers used viscosity ratios differing by 2-3 orders of magnitude. When the viscosity of the wall is 3 orders of magnitude or larger than the viscosity of the convecting fluid, then, the deformation of the wall would be negligible. Importantly, even if the initial boundary is assumed perfectly straight, time integration of the mechanical solution allows for conclusions to be drawn on the deformation of the strong lid.

The widespread use of fixed boundary conditions does have historical roots in fluid dynamics models developed for air and water.  In geological situations we should be a lot more careful.  It is easy to "assume" viscosity contrasts of 3 orders of magnitude, but the rocks in the Himalayan subduction channel were pretty much the same as those outside it – mainly metasedimentary gneisses and schists.  More critically, these rocks did not behave as Newtonian fluids; they deformed predominantly by dislocation creep (Law et al., 2013; Waters et al., 2018), which shows a power-law relationship between strain-rate and stress, with a stress exponent $n$ usually taken to be between 3 and 5.  The viscosity is therefore a function of the stress;  for $n = 3$, doubling the stress results in a drop in viscosity by a factor of 4.  In the model illustrated in Figure 2 of M2018, the shear strain rate in the upper part of the channel is $\sim 6 \times 10^{-14}$ s$^{-1}$ so if the viscosity is $10^{21}$ Pa s, the shear stress will be $\sim 60$ MPa. If the channel exerts a dynamic overpressure of 1.5 GPa on the upper plate, the shear stress in the upper plate will increase to $\sim 1.5$ GPa.  For power law creep with $n = 3$, this will reduce the viscosity of the upper plate by a factor of 600, which makes a nonsense of the assumption of M2018 that it has a viscosity 3 orders of magnitude larger than that in the channel.

P6  "If a dynamic overpressure of 1.5 GPa is applied from below to the upper boundary of the channel, a physical mechanism is required that is capable of keeping the boundary fixed, and

M2018 give no indication what this might be." This statement is not true. Marques and co-workers clearly state that this can occur if the walls are strong, so that the boundary would behave as if the lid were rigid. It is the high viscosity of the channel wall (that can build up large stresses) that is responsible for keeping the boundary nearly fixed.

The point that neither M2018 nor Moulas consider is that even if we assume the walls have a high enough viscosity to confine the overpressure, the load will still produce an elastic response. We know the elastic properties of rocks in general fairly well, and we have geophysical estimates of the effective elastic thickness in southern Tibet, so we can calculate that response. The results, as I show, are clearly incompatible with what we observe.

P7 "the only load acting downwards on the upper boundary is the lithostatic pressure"
This statement cannot be true for a deforming lithosphere with topography and density changes. It's close enough. Technically, it's only true on horizontal length scales larger than the scales of uncompensated topographic and density contrasts, which given the low effective elastic thickness in the region, means a few tens of km at most (note that the Himalayas as a whole are supported by the elastic strength of the underthrust Indian plate, but that doesn't affect our arguments about the upper plate). M2018 assume the upper plate is not deforming, but it could still sustain significant tectonic overpressure due to deviatoric stress. On that point M2018 have another problem, however: the shear stress exerted by the return flow in the channel tends to drag the upper plate up the dip of the subduction zone. Force balance then requires a tensile deviatoric normal stress in the upper plate parallel to the dip of the subduction zone, so the minimum principal stress will be ~ horizontal, and the tectonic overpressure will be negative. As a result, the load exerted by the upper plate will actually be less than lithostatic.

P8 "In the case of a subduction channel, the configuration can be approximated by the analysis for flexural doming above an igneous intrusion presented by Turcotte & Schubert (2002)."
The applicability of this solution in the subduction channel is not entirely justified. An important assumption for the application of the solution of Turcotte and Schubert (2002) is that the initial condition is known. Firstly, the layers of rocks are assumed to be horizontal and secondly, the deflections are calculated from this initial stage. By contrast, the configuration of Marques and co-workers is not an initial condition. Their solution is meant to depict the current configuration that satisfies force balance. Therefore, the plate deflection evolution in the Marques et al (2018) viscous model must be integrated over time as it is commonly done in Geodynamic modelling of slow viscous flow e.g. Gerya (2010).

To be frank, I find this comment a bit silly. M2018 do not discuss the time evolution of their model, and because they assume the upper plate is rigid, it won't change its geometry with time anyway. So there is no point to answer here.

P9. "Various lines of evidence suggest than an upper limit of ~120 MPa shear stress is reasonable for continental lithosphere in actively deforming regions".
These values for shear stress have no universal applicability. There are numerous models that use experimentally determined flow laws that would not agree with such a statement. Stresses on the order of 100MPa are the minimum required to support topography in mountainous regions only if the entire lithosphere is stressed in a uniform manner (average stresses). Clearly, this is highly

improbable since the presence of viscosity heterogeneities would result to regions in which the shear stress would be significantly higher or lower (Schmalholz et al., 2018, 2014).

I discuss this issue in some detail in my response to Referee 1. Mechanical data on dry diabase or granulite is not really relevant to a mountain range made up largely of mica schist and quartz-feldspar-biotite gneiss.

P10. "The model set-up by M2018 does not conform with these important principles …."
As mentioned in my previous review, instead of comparing the results of the Marques et al. (2018) model with natural observations, Prof. Platt compares the results of his own elastic flexure model with natural observations. However, the assumptions lying behind the elastic flexure formula are different to those invoked by Marques and co-authors (2018) in their viscous-flow model.

It may be reasonable to assume Newtonian viscous flow when modeling a subduction channel. My discussion concerns the behaviour of the upper plate, which provides the upper boundary condition for the model. Real rocks do exhibit elastic behaviour, and this can coexist with a viscous response, as in Maxwell and Bingham viscoelasticity. The elastic response provides a useful way to test the conclusions of the channel flow model in this case: the model fails spectacularly.

[revised manuscript text omitted]